# Global 1km Land Surface Parameters for Kilometer-Scale Earth System Modeling

2        Lingcheng Li, Gautam Bisht, Dalei Hao, L. Ruby Leung

Atmospheric, Climate, and Earth Sciences Division, Pacific Northwest National Laboratory,
Richland, WA, USA
Correspondence: Lingcheng Li (lingcheng.li@pnnl.gov) and Gautam Bisht
(gautam.bisht@pnnl.gov)
**Abstract**
Earth system models (ESMs) are progressively advancing towards the kilometer scale (k-scale).
However, the surface parameters for Land Surface Models (LSMs) within ESMs running at the k-
scale are typically derived from coarse resolution and outdated datasets. This study aims to develop
a new set of global land surface parameters with a resolution of 1 km for multiple years from 2001
to 2020, utilizing the latest and most accurate available datasets. Specifically, the datasets consist
of parameters related to land use and land cover, vegetation, soil, and topography. Differences
between the newly developed 1k land surface parameters and conventional parameters emphasize
their potential for higher accuracy due to the incorporation of the most advanced and latest data
sources. To demonstrate the capability of these new parameters, we conducted 1 km resolution
simulations using the E3SM Land Model version 2 (ELM2) over the contiguous United States.
Our results demonstrate that land surface parameters contribute to significant spatial heterogeneity
in ELM2 simulations of soil moisture, latent heat, emitted longwave radiation, and absorbed
shortwave radiation. On average, about 31% to 54% of spatial information is lost by upscaling the
1 km ELM2 simulations to a 12 km resolution. Using eXplainable Machine Learning (XML)
methods, the influential factors driving the spatial variability and spatial information loss of ELM2
simulations were identified, highlighting the substantial impact of the spatial variability and
information loss of various land surface parameters, as well as the mean climate conditions. The
comparison against four benchmark datasets indicates that ELM generally performs well in
simulating soil moisture and surface energy fluxes. The new land surface parameters are tailored
to meet the emerging needs of k-scale LSMs and ESMs modeling with significant implications for
advancing our understanding of water, carbon, and energy cycles under global change. The 1 km
land surface parameters are publicly available at https://zenodo.org/records/10815170 (Li et al.,

32    2024).

## 1. Introduction

Aided by advancements in computing power, it has become increasingly feasible to run land surface models (LSMs) and Earth system models (ESMs) at the kilometer scale (k-scale) to improve our understanding of Earth system processes. The emergence of k-scale modeling has the potential to improve the accuracy of climate simulations significantly and allow for explicit modeling of physical processes that were previously poorly represented in climate models (Nat. Clim. Chang. 2022), such as modeling of mesoscale convective systems in the atmosphere (Slingo et al., 2022) and mesoscale eddies in ocean (Hewitt et al., 2022). Simultaneously, land modeling has also witnessed a surge of interest in hyper-resolution modeling, initially proposed by Wood et al. (2011), which aims to model land surface processes at a horizontal resolution of 1 km globally and 100 m or finer for continental or regional domains. The motivation behind hyper-resolution modeling is to address the requirements of operational forecasting like extreme events, and to enhance our understanding of hydrological and biogeochemical cycling, and land–atmosphere interactions. High-resolution LSMs have been increasingly applied in various fields, as demonstrated by recent examples, such as 30-meter soil moisture simulations over the contiguous United States (CONUS) (Vergopolan et al., 2020, 2021, 2022), 500-meter hyper-resolution modeling of surface and root zone soil moisture over Oklahoma (Rouf et al., 2021), 1-km simulations over Southwestern US (Singh et al., 2015), 3-km simulations over eastern Tibetan Plateau to understand hydrological changes over mountainous regions (Yuan et al., 2018; Ji and Yuan, 2018), 6-km simulations over China to reduce simulations errors of hydrological variables (Ji et al., 2023). High-resolution modeling can better capture the land surface heterogeneity and could improve simulations of terrestrial water and energy cycles (Giorgi and Avissar, 1997; Chaney et al., 2018; Xu et al., 2023), biogeochemical cycles (Chaney et al., 2018), as well as land–

atmosphere coupling (Liu et al., 2017; Zhou et al., 2019; Bou-Zeid et al., 2020). For example,
Singh et al. (2015) demonstrated that increasingly capturing topography and soil texture
heterogeneity at finer resolutions (e.g., 1 km) improves land surface modeling of water and energy
variables. Li et al. (2022) have shown that the spatial heterogeneities of land surface parameters
(including land use and land cover (LULC) and topography) are essential for modeling the spatial
variability of land surface energy and water partitioning. Hao et al. (2022) found that 1 km
simulations with sub-grid topographic configurations can better capture the topographic effects on
surface fluxes.

The parameters for LSMs within ESMs being run at the k-scale are typically derived from coarse
resolution datasets or outdated datasets. Consequently, k-scale modeling may not accurately
represent fine-scale land surface heterogeneity unless high-resolution land surface parameters at
the kilometer or finer scales are utilized. Publicly available land surface parameters are primarily
provided at coarse resolutions and based on outdated datasets (see details in Table 1). For example,
the Community Land Model version 5 (CLM5; Lawrence et al., 2019) typically relies on land
surface parameters with spatial resolutions ranging from 1km to 0.5º based on source datasets that
were processed more than 10 years ago (see Table 1 for details). Although LULC-related
parameters are available at a relatively high resolution of 0.05º, they are temporally static and were
derived from a combination of data from different years spanning 1993 to 2012 (Table 1). Leaf
area index (LAI) was derived from the now outdated products of Moderate Resolution Imaging
Spectroradiometer (MODIS) collection 4 (Myneni et al., 2002). The canopy height for tree Plant
Functional Types (PFTs) is based on forest canopy height data derived from the Geoscience Laser
Altimeter System (GLAS) aboard ICESat, collected in 2005 (Simard et al., 2011). Canopy height
for short vegetation is represented by PFT-specific values that remain invariant in space (Bonan et
al., 2002). Soil sand and clay content were obtained from the International Geosphere-Biosphere
Programme (IGBP) soil dataset (Global Soil Data Task 2000) consisting of 4931 soil mapping
units (IGBP, 2000). These CLM5 land surface parameters have been widely utilized in the LSMs
and ESMs communities, despite being developed over a decade ago. Subsequently, Ke et al. (2012;
hereafter referred to as K2012) developed an updated set of LULC and vegetation-related land
surface parameters for CLM4 at a resolution of 0.05º. These parameters were developed based on
MODIS collection 5 products or datasets derived from MODIS collection 5 products, including
PFTs and non-vegetation land cover, LAI, and Stem Area Index (SAI). K2012 has also been widely
used by LSMs, including CLM (e.g., Leng et al., 2013; Ke et al., 2013; Singh et al., 2015; Xia et
al., 2017) and the Energy Exascale Earth System Model (E3SM) Land Model (ELM) (e.g.,
Caldwell et al., 2019; Leung et al., 2020; Li et al., 2022). However, the CLM5 and K2012 datasets,
with their relatively coarse resolution and reliance on outdated data from over a decade ago, may
not fully meet the requirements for k-scale modeling. Additionally, these datasets include LULC,
LAI, and SAI that are year invariant. Consequently, they are inappropriate for studies involving
LULC changes, such as urbanization. In addition, some recently developed land surface processes
and their associated parameters are not included in previous datasets. For instance, Hao et al. (2021)
introduced a sub-grid topographic parameterization of solar radiation with five associated
topographic factors in ELM, which have been found to significantly affect the surface energy
budget. the surface energy budget.

High-resolution and up-to-date datasets at kilometer or finer resolutions are now widely available
and can be utilized to derive more accurate land surface parameters for k-scale LSM simulations.
For example, the MODIS Land Cover Type Collection 6 (MCD12Q1 C6) data product provides
global land cover types yearly from 2001 to the present (Friedl et al., 2019; Sulla-Menashe et al.,
2019) at 500-meter resolution. Compared to the MODIS Collection 4 (used in CLM5 land surface
parameters) and Collection 5 products (used in K2012 land surface parameters), the C6 data
represents a significant advancement in algorithm improvements and the quality of land cover
information. Despite the availability of high-resolution MODIS LAI products, such as the 500 m
MCD15A2H (Myneni et al., 2021), they suffer from noise and gaps with spatially and temporally
inconsistent values due to clouds, seasonal snow cover, instrument issues, and uncertainties in
retrieval algorithms (Yuan et al., 2011). To address these limitations, Yuan et al. (2011)
reprocessed MODIS LAI products and generated a more accurate and spatiotemporally continuous
and consistent LAI dataset that is available continuously to the present period. Additional high-
resolution and up-to-date datasets are available for preparing land surface parameters, such as soil
texture and soil organic matter at 250-meter resolution (Poggio et al., 2021) and vegetation height
at 10-m resolution (Lang et al., 2023).

This study aims to develop a new set of global land surface parameters with a resolution of 1 km
for multiple years, utilizing the latest and most accurate available datasets. These parameters will
be tailored to meet the needs of k-scale Earth system modeling. The newly developed land surface
parameters include four categories: (1) LULC-related parameters, such as the spatial distributions
of PFTs, lakes, wetlands, urban areas, and glaciers; (2) vegetation-related parameters, including
PFTs' LAI and SAI for multiple years ranging from 2001 to 2021, and the canopy top and bottom
height; (3) soil-related parameters, such as soil textures and soil organic matter; and (4)
topography-related parameters, such as elevation, slope, aspect, and sub-grid topographic factors.
We conducted a comparison of the new 1k parameters against the K2012 and ELM2/CLM5 default
parameters. Utilizing ELM version 2 (ELM2) as a testbed, we demonstrated the modeling
capability enabled by the new high-resolution parameters through a 5-year simulation at 1 km
resolution over the CONUS. We performed a spatial scaling analysis on four ELM2 simulated
variables, which included soil moisture, latent heat, emitted longwave radiation, and absorbed
shortwave radiation, to underscore the significance of high-resolution land surface parameters on
ELM2 simulations. We employed eXplainable Machine Learning (XML) methods to evaluate the
most important factors of land surface parameters and climate conditions (e.g., mean temperature
and precipitation) in driving the spatial variability and spatial information loss of ELM2
simulations.

**2. Development of 1km land surface parameters**

In this study, all the land surface parameters were developed globally at a resolution of approximately 1 km (i.e., 1/120°, hereafter referred to as 1 km; Table 1). The LULC-related parameters, soil properties, canopy height, and elevation were processed via Google Earth Engine (GEE; Gorelick et al., 2017). The LAI was processed using an area-weighted average from its original 450 m resolution obtained from Beijing Normal University (Yuan et al., 2011). All data sources utilized in this study have been rigorously validated in their respective original publications. The detailed methods for deriving these parameters are described below.

Table 1 Comparison between new and previous land surface parameters

| Category | Land surface parameters | This study | ELM2 / CLM5 * | K2012 |
|---|---|---|---|---|
| LULC | PFTs, Lake, Glacier, Urban | • Resolution: 1 km, yearly, 2001-2020<br><br>• Data source: 500 m, yearly, MODIS collection 6 (Friedl et al., 2019) | • Resolution: 0.05°, temporally static, processed based on data from mixed years<br><br>• PFTs data source: mixed years from 1993 to 2001; 500 m, MODIS Vegetation Continuous Fields (Hansen et al., 2003); 1 km, tree cover (Defries et al., 2000); 10 km (5 arc minutes), cropland (Ramankutty and Foley, 1999); 1 km, MODIS land cover collection 4 (Friedl et al., 2002)<br><br>• Lake data source: 3 km (90 arc seconds) lake data (Kourzeneva 2009, 2010)<br>• Glacier data source: glacier and ice sheet vector data (Arendt et al. 2012; Rastner et al. 2012)<br>• Urban data source: 1 km urban data (Jackson et al., 2010) | • Resolution: 0.05°, year 2005<br><br>• Data source: 500 m, yearly, MODIS collection 5 (Friedl et al., 2010) |
| Vegetation | LAI, SAI | • Resolution: 1 km, monthly, 2001-2020<br><br>• Data source: 450 m, 8-day, reprocessed MODIS collection 6 LAI (Yuan et al., 2011; Friedl et al., 2019) | • Resolution: 0.5°, 12 months<br><br>• Data source: 1 km, 8-day, MODIS collection 4 LAI (Myneni et al., 2002) | • Resolution: 0.05°, year 2005<br><br>• Data source: 450 m, 8-day, reprocessed MODIS collection 5 LAI (Yuan et al., 2011; Friedl et al., 2010) |
| | Canopy top height, Canopy bottom height | • Resolution: 1 km, temporally static<br><br>• Data source: 10 m, vegetation canopy height (Lang et al., 2023) | • Resolution: 0.5° or PFT specified value, temporally static<br><br>• Tree PFT data source: 1 km, forest canopy height derived using 2005 GLAS aboard ICESat data (Simard et al., 2011);<br><br>• Short vegetation data source: PFT specific values (Bonan et al., 2002) | -- |
| Soil | Percent sand, Percent clay | • Resolution: 1 km, temporally static<br><br>• Data source: 250 m, Soilgrid v2 (Poggio et al., 2021) | • Resolution: 10 km (0.083°), temporally static<br><br>• Data source: IGBP soil data of 4931 mapping units (IGBP, 2000) | -- |
| | Soil organic matter | | | |
| Topography | Elevation, Slope, Standard deviation of elevation | • Resolution: 1 km, temporally static<br><br>• Data source: 90 m, MERIT Hydro elevation (Yamazaki et al., 2019) | • Resolution: merge of 1 km and 10 arc minutes, temporally static<br><br>• Data source: global most regions are based on USGS HYDRO1k (Verdin and Greenlee 1996); but 10 arc minute data is used over Greenland and Antarctica. | -- |
| | Aspect, Sky view factor, Terrain configuration factor | • Resolution: 1 km, temporally static<br><br>• Data source: 90 m, MERIT Hydro elevation (Yamazaki et al., 2019) | -- | |

* ELM2 and CLM5 share the same default land surface parameters, detailed descriptions available at:
https://escomp.github.io/ctsm-docs/versions/release-clm5.0/html/tech_note/index.html.

**2.1 LULC-related parameters**

In this study, the MODIS MCD12Q1 version 6 (Friedl et al., 2022) was employed to ascertain the Plant Functional Types (PFT) as well as other non-vegetative land categories at a spatial resolution of 1 km spanning the years 2001 to 2020. The integrity of the MODIS land cover product has been established through a 10-fold cross-validation accuracy assessment using the Terrestrial Ecosystem Parameterization database (Sulla-Menashe et al., 2019). This land cover product offers richer and more flexible land cover data with higher accuracy and substantially less year-to-year stochastic variation in classification results (Sulla-Menashe et al., 2019). Being the sole operational global land cover product available with annual intervals, it addresses a significant gap in the realm of global change research.

The original MODIS land cover data was first resampled to 1 km from its original 500 m resolution using a majority resampling method in GEE. At such a high 1km resolution, we did not consider the proportion of different land cover types within each grid. Instead, we assigned 100% of a grid cell to the major land cover type. Specifically, the MCD12Q1 LC_Type 5 PFT classification layer was used to determine the distributions of the seven PFTs, as well as lake, urban, and glacier, following the method outlined in Ke et al. (2012) and summarized below:

- The seven PFTs include needleleaf evergreen trees, needleleaf deciduous trees, broadleaf evergreen trees, broadleaf deciduous trees, shrub, grass, and crop. These PFTs were further reclassified into 15 categories (Table S1) that are typically used in LSMs based on the rules presented in Bonan et al. (2002a) with the assistance of 1 km precipitation and surface air temperature from WorldClim V1 (Hijmans et al., 2005).

• Grass was reclassified as C3 and C4 grass using the approach presented by Still et al. (2003),
with the assistance of monthly LAI (processed in section 2.2.1) and meteorological
variables from WorldClim V1.
• The "non-vegetated land" was classified as barren soil class.
• The "permanent snow and ice" was assigned as the glacier land unit.
• Global lakes were identified based on the classification of "water bodies" over the global
land, constrained using the global land mask obtained from Natural Earth
(https://www.naturalearthdata.com/).
• The urban land unit was determined based on the MODIS "urban and built-up"
classification. These urban grids were further classified into three urban classes, namely,
tall building district (TBD), high density (HD), and medium density (MD), based on
Jackson et al. (2010; hereinafter referred to as J2010). J2010 generated global urban extent
maps for the TBD, HD, and MD classes at a spatial resolution of 1 km, based on rules of
building height and vegetation coverage fraction
(https://gdex.ucar.edu/dataset/188a_oleson/file.html). However, the J2010 dataset is
temporally static and cannot reflect changes in urban boundaries over time. Therefore, we
reclassified the yearly MODIS urban land class as TBD, HD, and MD based on the J2010
dataset using the nearest neighbor sampling method for each year.
After determining the distribution of 15 PFTs, bare soil, lake, glacier, and urban land, any
remaining 1 km grids were assigned as ocean (Table S1). It should be noted that the wetland land
unit was not explicitly classified in this study. This is because, instead of treating wetlands as an
individual land unit, many LSMs (e.g., ELM2 and CLM5) integrate wetland functioning processes
prognostically within other land units where a surface water storage component is implemented to
represent wetland functioning.

**2.2 Vegetation-related parameters**
**2.2.1 Monthly LAI and SAI**
The monthly LAI parameters were obtained from Beijing Normal University (BNU_LAI; Yuan et
al., 2011). BNU_LAI, an enhanced version of the MODIS LAI product, has been subjected to
thorough quality control, incorporating multiple algorithms for improved accuracy (Yuan et al.,
2011). Its validation involved an extensive array of LAI reference maps and employed the bottom-
up approach advocated by the CEOS Land Product Validation sub-group (Morisette et al., 2006).
Compared to the original MODIS LAI, the BNU_LAI dataset exhibits superior performance, along
with enhanced spatiotemporal continuity and consistency. The 8-day BNU_LAI product at a
resolution of 15 seconds (~450 m) over 2001–2020 was downloaded from
http://globalchange.bnu.edu.cn/research/laiv061. Subsequently, the data were resampled to a
resolution of 1 km using an area-weighted average method and averaged temporally for each
month. The processed monthly LAI at 1 km resolution was subsequently assigned to each of the
15 PFTs described above at each grid. The monthly SAI was then calculated based on the
processed monthly LAI using the methods and PFT parameters described in Zeng et al. (2002).

**2.2.2 Vegetation canopy height**
We leveraged a global vegetation canopy height dataset sourced from Lang et al. (2023). This
dataset, derived using a probabilistic deep learning model, fuses Sentinel-2 images with the Global
Ecosystem Dynamics Investigation (GEDI) to retrieve canopy height. It stands out as the inaugural
global canopy height dataset offering consistent, wall-to-wall coverage at a 10 m spatial resolution
across all vegetation types. Assessments using hold-out GEDI reference data and comparisons
with independent airborne LiDAR data demonstrate that the approach outlined by Lang et al. (2023)
produces a meticulously quality-controlled, state-of-the-art global map product, accompanied by
quantitative uncertainty estimates. The canopy height served as the canopy top height parameter.
Canopy bottom height was calculated by multiplying PFT-based ratios derived from the ratio of
ELM2's (same as CLM5) canopy top and bottom heights for different PFTs (Table S2).

**2.3 Soil-related parameters**
We obtained the Soilgrid v2 data with an original resolution of 250 m (Poggio et al., 2021) to
prepare soil properties. Soilgrid is generated using machine learning based on multiple data
sources of soil profiles and remote sensing data (Hengl et al., 2017). The soil product underwent
rigorous quantitative evaluation using a cross-validation method, which ensures alignment with
established pedo-landscape features and provides spatial uncertainty to guide product users
(Poggio et al., 2021). Soilgrid v2 provides percent clay, percent sand, and soil organic matter for
six standard soil layers: 0–5 cm, 5–15 cm, 15–30 cm, 30–60 cm, 60–100 cm, and 100–200 cm.
The original SoilGrid version 2 data obtained from GEE were processed at 1 km resolution with
multiple layers using an area-weighted average method. To facilitate the demonstration, we
restructured the six soil layers vertically into ELM2's ten effective soil layers (0–1.8 cm, 1.8–4.5
cm, 4.5–9.1 cm, 9.1–16.6 cm, 16.6–28.9 cm, 28.9–49.3 cm, 49.3–82.9 cm, 82.9–138.3 cm, 138.3–
229.6 cm, and 229.6–380.2 cm) using the nearest neighboring method. It should be noted that the
lake module in ELM2 and CLM5 requires soil properties, but the Soilgrid v2 data may not provide
coverage over water surfaces. To address this, we utilized the nearest neighbor sampling method
to map the 1 km soil properties onto the terrestrial water surface.

**2.4 Topography-related parameters**
We employed the digital elevation from the Multi-Error-Removed Improved-Terrain DEM
(MERIT DEM, Yamazaki et al., 2019) to obtain topography-related parameters. The MERIT DEM
provides globally consistent elevation data at 90 m resolution, distinguished by its exceptional
vertical accuracy. This accuracy was rigorously validated against ICESat's lowest elevations in
both forested and non-forested regions and was further benchmarked using the UK's premium
airborne LiDAR DEM (Yamazaki et al., 2019). We first acquired the 1km elevation and standard
deviation of elevation using GEE based on the original 90 m elevation. Further, we calculated the
slope, aspect, sky view factor, and terrain configuration factor from the 1km elevation using the
parallel computing tool developed by Dozier (2022). The sky view factor represents the proportion
of visible sky limited by adjacent terrain, and the terrain configuration factor describes the
proportion of adjacent terrain which is visible to the ground target. Finally, to drive the
parameterization of sub-grid topographical effects on solar radiation (Hao et al., 2022) in ELM2,
we calculated the $\sin(slope) \cdot \sin(aspect)$ and $\sin(slope) \cdot \cos(aspect)$ for calculating the
local solar incident angle, and two normalized angle-related factors, the sky view factor, and terrain
configuration factor by $\cos(slope)$. It is important to note that the standard deviation of elevation
calculated in this study is specific to the 1 km resolution simulation. For applications requiring
coarser resolutions (e.g., 0.5 degree), the standard deviation should be recalculated directly from
the 1 km elevation, rather than averaging from the 1k standard deviation of elevation.

**2.5 Comparison between new and existing land surface parameters**

In this study, since the data sources used to develop the 1k global land surface parameters have already undergone rigorous validation, we do not perform additional evaluations against reference datasets (e.g., observations). Instead, our focus is on comparing the newly developed 1k parameters with those from K2012 and the ELM2/CLM5 default parameters. The K2012 parameters, obtained through personal communication (refer to the data availability section for details). The ELM2/CLM5 default parameters were sourced from the CESM input data repository (https://svn-ccsm-inputdata.cgd.ucar.edu/trunk/inputdata/). Given the different resolutions of these datasets—our new parameters at 1km, K2012 at 0.05 degree, and ELM2/CLM5 defaults with varying resolutions—we adapt our comparison at different resolutions for different variables.

For PFT parameters, we aggregated both the 1k new parameters and the 0.05-degree K2012 data to the 0.5-degree resolution of the ELM2/CLM5 default. For non-vegetated land units (i.e., urban, glacier, and lake), we upscaled the 1k new parameters to a 0.05-degree resolution to align with the ELM2/CLM5 default. It is important to note that the urban parameter in K2012 is only available for the northern hemisphere, due to limitations in data acquisition.

When comparing LAI, we aggregated the 1k new and K2012 LAI to 0.5-degree resolution, matching the ELM2/CLM5 default LAI/SAI resolution. We excluded the comparison of SAI from our analysis due to the limited availability of the global K2012 dataset, from which we only acquired coverage for North America. We have not included a comparison of vegetation canopy height (top and bottom parameters) in our study. This is because the K2012 dataset does not contain these parameters, and the ELM2/CLM5 default parameters in the CESM input data repository provide only tabular values for each PFT, rather than spatially variable canopy heights for tree PFTs.

For soil and topography-related parameters, our comparison was limited to the 1k new parameters
and the ELM2/CLM5 default, as K2012 does not include these parameters. Specifically, for soil
comparisons, we aggregated the new 1k parameters to 0.083° resolution to match the ELM2/CLM5
default soil parameters. For topography, given that the ELM2/CLM5 default parameters is a
combination of 1k and 10 arc-minute data sources, we simplify the comparison by aggregating
both the new 1k parameters and ELM2/CLM5 default to 0.5-degree resolution, including elevation
and slope.

**3. K-scale demonstration simulation over CONUS**

**3.1 Experiment design**

To demonstrate the capability of 1 km datasets, we conducted ELM2 simulations over CONUS at the resolution of 1 km, using the newly developed 1 km land surface parameters for 2010. We used atmospheric forcing from the Global Soil Wetness Project Phase 3 (GSWP3; Kim, 2017) with a spatial resolution of 0.5° to drive ELM. The spatial homogeneity of atmospheric forcings within 0.5° grid cell guarantees that the spatial variability of ELM simulated variables (e.g., latent heat) within 0.5° grid cell is solely attributable to the heterogeneity of the 1 km land surface parameters. There are approximately 12 million effective grids over CONUS. We ran ELM for five years (2010–2014), and the last year's simulation was used for analysis. We specifically analyzed the annual mean of surface layer soil moisture (SM, $m^3/m^3$), latent heat (LH, $W/m^2$), emitted longwave radiation (ELR, $W/m^2$), and absorbed shortwave radiation (ASR, $W/m^2$).

**3.2 Spatial scaling analysis**

We conducted a spatial scaling analysis following the method described in Vergopolan et al. (2022) on the 1 km ELM simulation data to better understand how k-scale spatial heterogeneity in the four ELM-simulated variables (mentioned in Section 3.1) induced only by spatial heterogeneity of land surface parameters changes across spatial scales. First, we performed upscaling by averaging the 1 km (=1/120°) land surface parameters and the four ELM-simulated variables to coarser spatial scales, $\lambda_{scale}$ of 1/60°, 1/40°, 1/30°, 1/24°, 1/20°, and 1/10°, and calculated the spatial standard deviation ($\sigma_{scale}$) within each 0.5° × 0.5° box at each spatial scale (Table 2). Second, we quantified the changes in spatial variability at different spatial scales compared to the original 1km resolution by calculating the ratio of $\sigma_{scale}$ to $\sigma_{1\,km}$. Third, we fitted a $\log\left(\frac{\sigma_{scale}}{\sigma_{1\,km}}\right) \propto$ $\beta \times \log\left(\frac{\lambda_{scale}}{\lambda_{1\,km}}\right)$ relationship, where $\beta$ is an indicator to quantify data spatial variability persistence

across scales (Hu et al., 1997). A more negative $\beta$ indicates a larger dependency of data spatial
variability on spatial scales, resulting in a higher information loss, denoted as $\gamma_{scale} =$
$(1 - \sigma_{scale}/\sigma_{1\,km}) \times 100\%$. In this study, we focus on information loss at a 12 km scale, denoted
as $\gamma_{12\,km}$. For simplicity in subsequent discussion, $\gamma_{12\,km}$ will be referred to as $\gamma$ in the results
section. Given the possibility that $\beta$ may not demonstrate significant temporal variation (Mälicke
et al., 2020), and considering that our scaling analysis is intended for demonstration purposes, our
spatial scaling analysis is based on the annual mean of ELM2 simulations.
It is crucial to clarify that the upscaled 1 km simulation results in the spatial scaling analysis are
not equivalent to the results obtained from a coarse resolution ELM conducted using upscaled
parameters. The spatial scaling analysis is intended to emphasize the value of high-resolution
modeling in capturing fine-scale spatial variabilities, and to highlight the contributions of high-
resolution land surface parameters on the simulated variables.

327                    Table 2. Spatial resolution and pixel number at different spatial scales.

| $\lambda_{scale}/\lambda_{1\,km}$ | 1 | 2 | 3 | 4 | 5 | 6 | 12 |
|---|---|---|---|---|---|---|---|
| Spatial resolution | 1km (1/120°) | 2km (1/60°) | 3km (1/40°) | 4km (1/30°) | 5km (1/24°) | 6km (1/20°) | 12km (1/10°) |
| Pixel number within 0.5° × 0.5° box | 60 × 60 | 30 × 30 | 20 × 20 | 15 × 15 | 12 × 12 | 10 × 10 | 5 × 5 |


**3.3 Attribution analysis utilizing XML methods**
We conducted additional analysis to determine the primary land surface parameters that influence
the spatial scaling of ELM simulations. We employed XML methods, specifically the eXtreme
Gradient Boosting (XGBoost; Chen and Guestrin, 2016) machine learning algorithm and the game
theoretic approach SHapley Additive exPlanations (SHAP; Lundberg and Lee, 2017; Lundberg et
al., 2018, 2020). XML methods were utilized to assess the influence of land surface parameters on
the spatial variability and information loss of ELM2 simulations across the CONUS. Taking spatial
variability as an example, we first computed the standard deviation ($\sigma$) within each 0.5º x 0.5º grid
for both 1 km resolution land surface parameters and simulations. Then, we train a machine
learning model to predict the spatial variability of each simulated variable (i.e., SM, LH, ELR,
ASR). We used the spatial variability (i.e., $\sigma$) and mean ($\mu$) of the land surface parameters and $\mu$
of precipitation and temperature as predictor variables, and the simulated variable's $\sigma$ as the target
variable. After training the machine learning model, we used SHAP to quantify the relative
importance and determine which factors were most important in driving the spatial variability of
the simulations. Similarly, we used this approach to identify the most critical drivers of information
loss.

**3.4 Reference datasets for evaluating ELM simulation**
We also performed a comparison of all four ELM-simulated variables against reference datasets.
It is important to note that we used the default model parameters and did not perform any
calibration (see discussions for details). For reference datasets, soil moisture was obtained from
the Global Land Evaporation Amsterdam Model (GLEAM; Martens et al., 2017), latent heat flux
data was from the MODIS product (Running et al., 2021), and both ELR and ASR data were
processed from the land component of the fifth generation of European ReAnalysis (ERA5_Land;
Muñoz-Sabater et al., 2021). For the soil moisture evaluation, we compared the surface layer soil
moisture from GLEAM (10 cm depth) with the weighted average of the first four-layer soil
moisture from ELM (about 11 cm depth).  To ensure comparability, we unified the spatial
resolution of both reference datasets and ELM simulations to a 0.5-degree resolution and focused
our analysis on the annual mean data for 2014.

## 4. Results

### 4.1 Demonstration of the global 1km land surface parameters

LAI generally shows high values in humid and warm regions, such as tropical rainforests, southeastern US, and southern Asia, and low values over arid or cold regions, such as central Australia, southwestern US, Middle East, Central Asia, and northern Canada (Figure 1a). At high resolution, the LAI dataset clearly reflects the detailed heterogeneity of vegetation distributions. In subregion R1 (Figure 1b), a relatively small LAI is distributed over mountain ridges and zero LAI over water surfaces (e.g., lakes). In subregion R2 (Figure 1c), the LAI pattern shows a large proportion of forest fragmentation caused by deforestation. In subregion R3 (Figure 1d), the LAI shows the distribution of agricultural land along with the river, river mouth, and lakes under an arid climate. R4 shows how urbanization affects vegetation distributions (Figure 1e).

Figure 2 demonstrates the distribution of plant functional types and other non-vegetation land units. High-resolution LULC types over multiple years can benefit studies related to LULC changes like urbanization and deforestation. Canopy height generally follows a similar spatial pattern with LAI, with high values in humid and warm regions and low values over arid or cold regions (Figure 3a). The percent clay shows high values over Southeast Asia, India, central Africa, and southeast South America, and low content over North Europe, South Africa and Alaska (Figure 3b). The topography factors follow the elevation patterns (Figures 3c and 3d), where there are large slopes and standard deviation of elevation over mountainous regions, such as the Rocky Mountains in North America, the Himalayas Mountains in Asia, and Andes Mountains in South America.

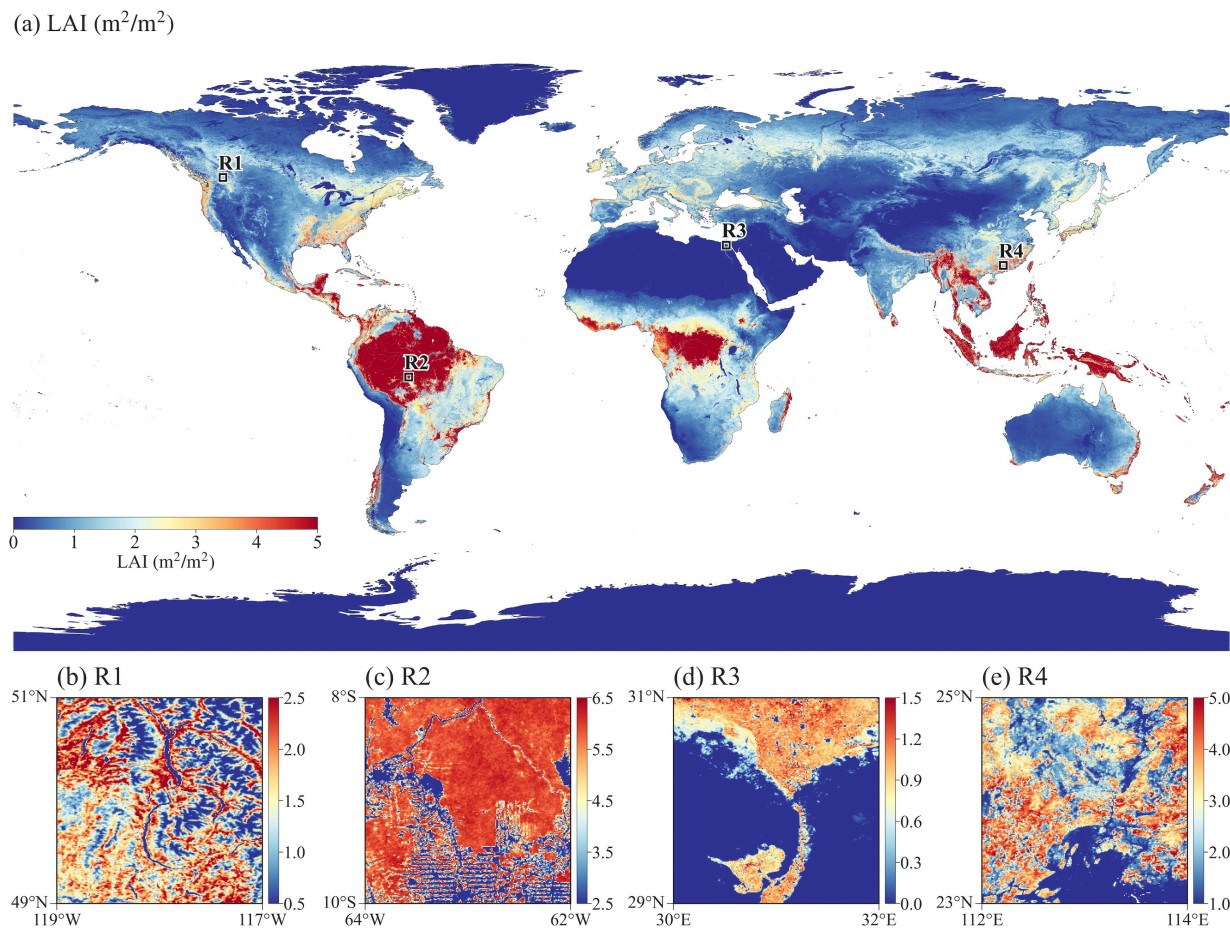


Figure 1. The spatial pattern of LAI (annual mean in 2010) over (a) global land and (b)~(e) four
subregions R1~R4 within 2-degree boxes marked in (a). Subregions R1~R4 represent
topography, deforestation, irrigations, and urbanization effects on LAI.

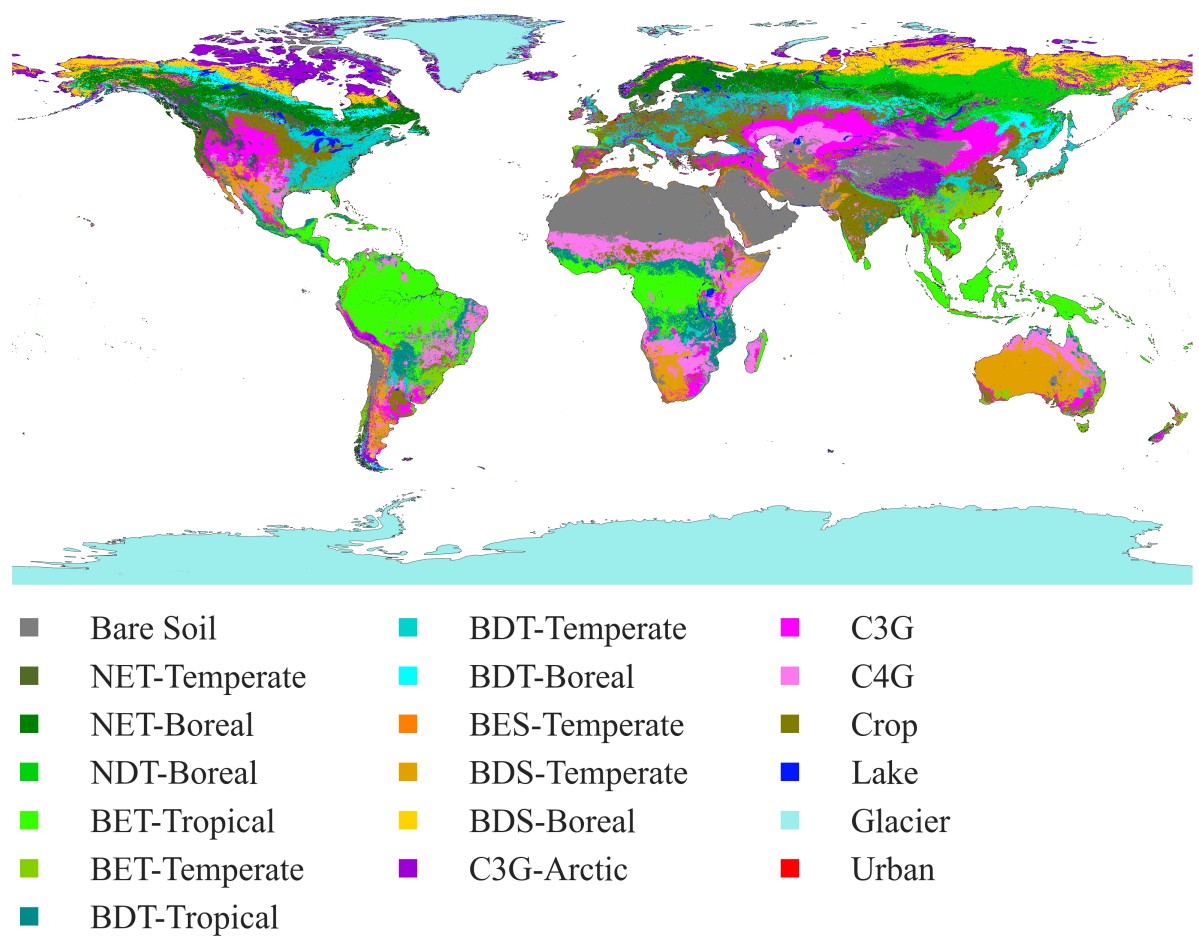


Figure 2. Global LULC distribution in year 2010. PFT abbreviations include: Bare Soil, Needleleaf

Evergreen Trees in temperate (NET-Temperate) and boreal (NET-Boreal) regions, Needleleaf

Deciduous Trees in boreal regions (NDT-Boreal), Broadleaf Evergreen Trees in tropical (BET-

Tropical) and temperate (BET-Temperate) regions, Broadleaf Deciduous Trees in tropical (BDT-

Tropical), temperate (BDT-Temperate), and boreal (BDT-Boreal) regions, Broadleaf Evergreen

Shrubs in temperate regions (BES-Temperate), Deciduous Shrubs in temperate (BDS-Temperate)

and boreal (BDS-Boreal) regions, C3 Grass in arctic (C3G-Arctic) and general (C3G) varieties,

C4 Grass (C4G), Crop, Lake, Glacier, and Urban.

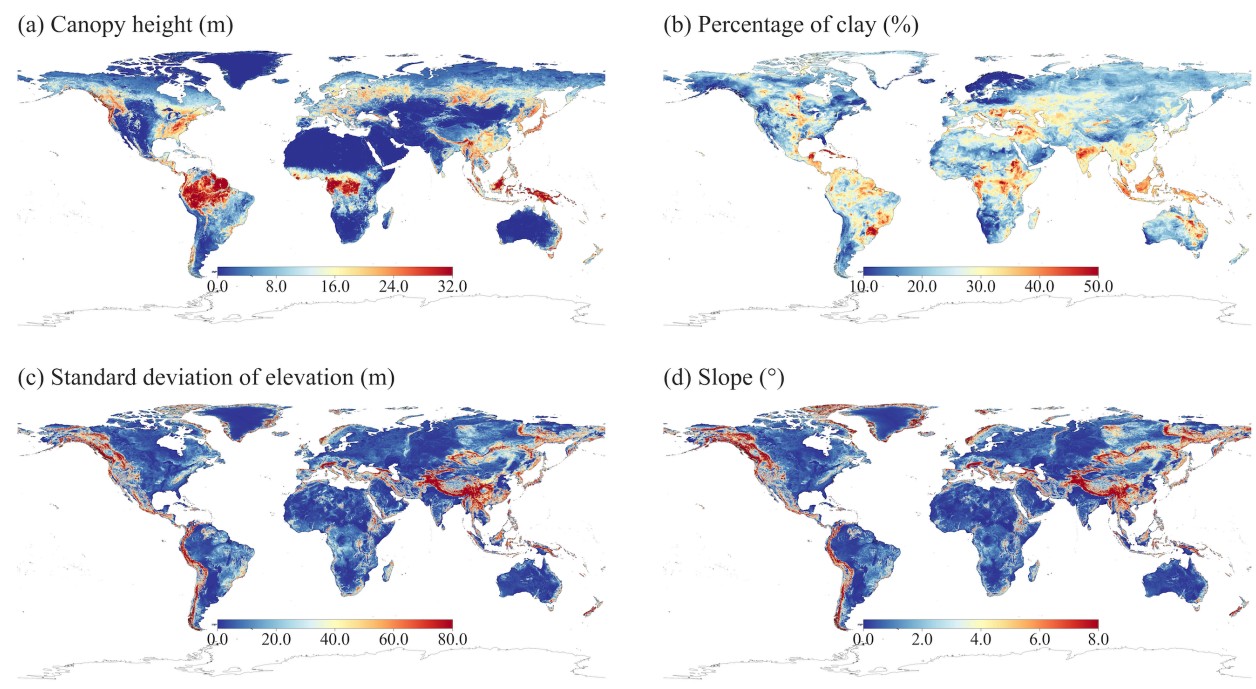

(a) Canopy height (m)

(b) Percentage of clay (%)

(c) Standard deviation of elevation (m)

(d) Slope (°)


Figure 3. Demonstration of global 1km datasets (a) Canopy top height, (b) percent clay, (c)

395                    standard deviation of elevation, and (d) slope.


## 4.2 Comparison between new and existing land surface parameters

The global distributions of different PFTs show varying degrees of difference when comparing the
new parameters with the K2012 and ELM2/CLM5 default parameters (Figure 4 and
Supplementary Figures S1 to S16). Predominant types such as bare soil, BET-Tropical tree, C3
and C4 grass, and crop are found consistently across all datasets. Notable differences include less
bare soil in the new parameters and K2012 compared to ELM2/CLM5 default, especially in high-
latitude North America, western US, South Africa, Central Asia, and Central Australia (Figure S1).
While the new NDT PFT shows larger coverage in Siberia than K2012 and ELM2/CLM5 (Figure
S4), BET-Tropical PFT is more prevalent in the new parameters across Central and South America
(Figure S5). BET-Temperate PFT has greater area coverage in southern China in the new
parameters (Figure S6). For BDT-Tropical, BDT-Temperate, and BDT-Boreal PFTs, both the new
and ELM2/CLM5 default parameters surpass K2012 data in coverage (Figures S7 to S9). The
coverage of new BDS-Temperate PFT is smaller than K2012 but larger than ELM2/CLM5 default
(Figure S11), and the new BDS-Boreal PFT is less extensive in the boreal northern hemisphere
compared to both K2012 and ELM2/CLM5 defaults (Figure S12). The C3-Arctic PFT shows
larger areas in the new parameters, particularly in northern Canada, with the new C4 grass PFT
being similar to that of K2012 and larger than ELM2/CLM5 C4 grass. Crop PFT is less extensive
in the new parameters, particularly in Southeastern China, Europe, South America, Africa, and
Australia.

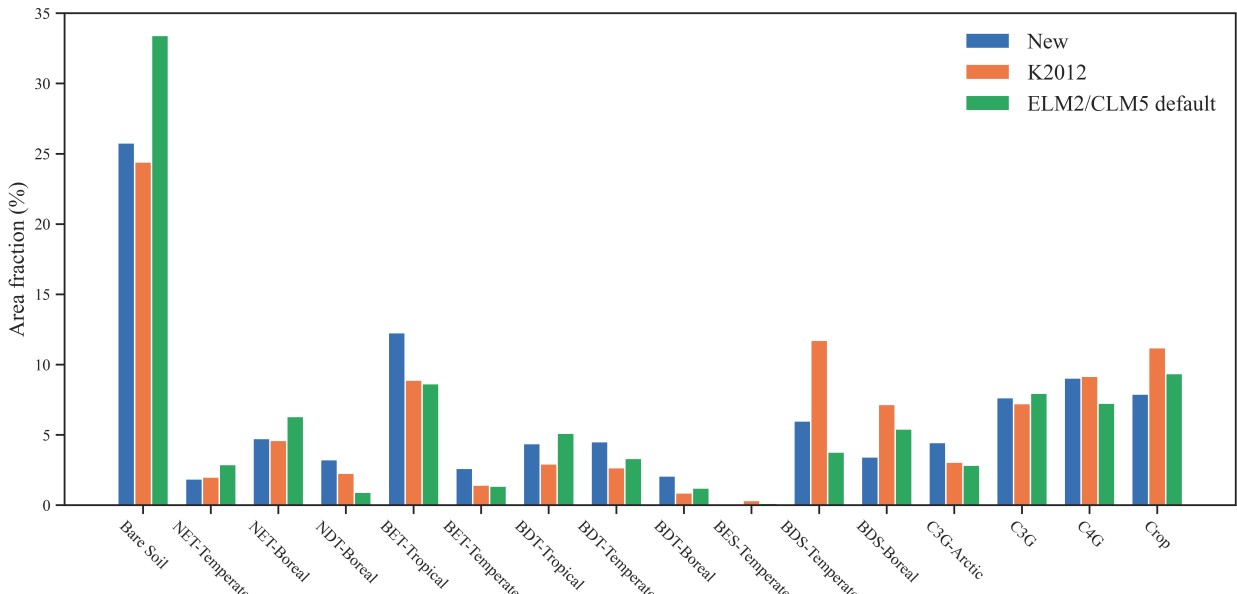


Figure 4. The global average area fractions of PFTs for three land surface parameter datasets. PFT
abbreviations used on the X-axis are displayed in Figure 2.

The global distributions of non-vegetated land covers of lake, glacier and urban areas vary among
the datasets (Figure S17–S19). The new dataset shows slightly less lake coverage than K2012, but
both are smaller than ELM2/CLM5 default, particularly in high-latitude North America (Figure
S17). Glacier coverage in the new parameter is around 0.7% smaller than K2012, with noticeable
differences in the Arctic North America, while ELM2/CLM5 default shows more extensive glacier
coverage in Antarctica (Figure S18). Regarding urban areas, K2012 has the smallest urban
coverage in the Northern Hemisphere compared to both the new dataset and ELM2/CLM5 default
(Figure S19). Meanwhile, ELM2/CLM5 default exhibits more expansive urban areas in India and
China than the new dataset and K2012.

The global annual mean LAI exhibits similar spatial patterns among the new parameter, K2012,
and ELM2/CLM5 (Figure 5). The overall global mean LAI for the new parameter (1.28 m²/m²) is
slightly higher than that of K2012 (1.14 m²/m²) and the ELM2/CLM5 default data (1.24 m²/m²).
In terms of spatial pattern, the new LAI, relative to K2012 (Figure S20a), shows lower values in
the NET-Boreal PFT over the northern hemisphere, but higher values in the BET-Tropical PFT
over the tropics. Similarly, compared with the ELM2/CLM5 default LAI (Figure S20b), the new
LAI also presents smaller values in both the NET-Boreal and NDT PFTs over the northern
hemisphere, but larger values in the BET-Tropical PFT regions.

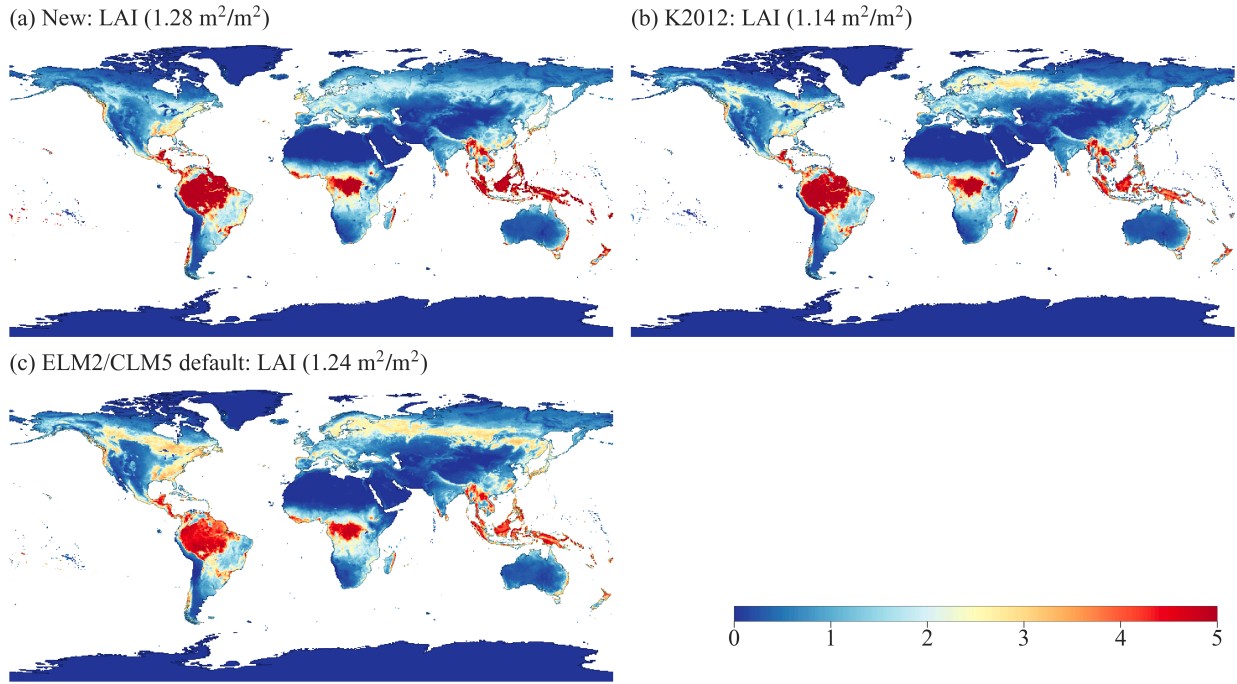

(a) New: LAI (1.28 m²/m²)

(b) K2012: LAI (1.14 m²/m²)

(c) ELM2/CLM5 default: LAI (1.24 m²/m²)

Figure 5. Comparison of global annual mean LAI for (a) new, (b) K2012, and (c) ELM2/CLM5 default parameters. The global average is indicated in the subplot title.

Soil parameters exhibit significant differences between the new and ELM2/CLM5 default datasets (Figures 6a-bc, S21, and S22). The global mean absolute differences between the new and ELM2/CLM5 default for percent sand, percent clay, and organic matter are 14.1%, 8.1%, and 30.5 kg/m³, respectively. Generally, the new soil parameters are spatially distributed more smoothly than those from ELM2/CLM5 with more patchy patterns (Figure 6a vs. 6b). Specifically, the new percent sand is higher in regions like Europe, Siberia, South Africa, and Southern Australia, but lower in areas such as the Lower Mississippi River Basin, North Africa, and Central and Southeastern Asia (Figure 6c). The new percent clay shows larger values in the Western US, North Africa, Central Asia, and Australia, but smaller values in Alaska and Eastern Europe (Figure S21).

For organic matter, the new parameter indicates smaller values in the Northern Hemisphere but
larger values in other global regions compared to the ELM2/CLM5 default (Figure S22).
Topography-related parameters exhibit broadly similar spatial patterns but with notable
differences between the new and ELM2/CLM5 default parameters, as seen in Figures 6d-6f and
S23. The new slope parameter generally shows a larger slope relative to the ELM2/CLM5 default,
particularly in mountainous regions (Figure 6f). This could be attributed to the new 1 km slope
being calculated from a finer 90 m resolution elevation. Differences in elevation between the new
and ELM2/CLM5 parameters are more pronounced in areas such as various mountainous regions,
Greenland, the Amazon Basin, the Tibetan Plateau, and Australia (Figure S23).

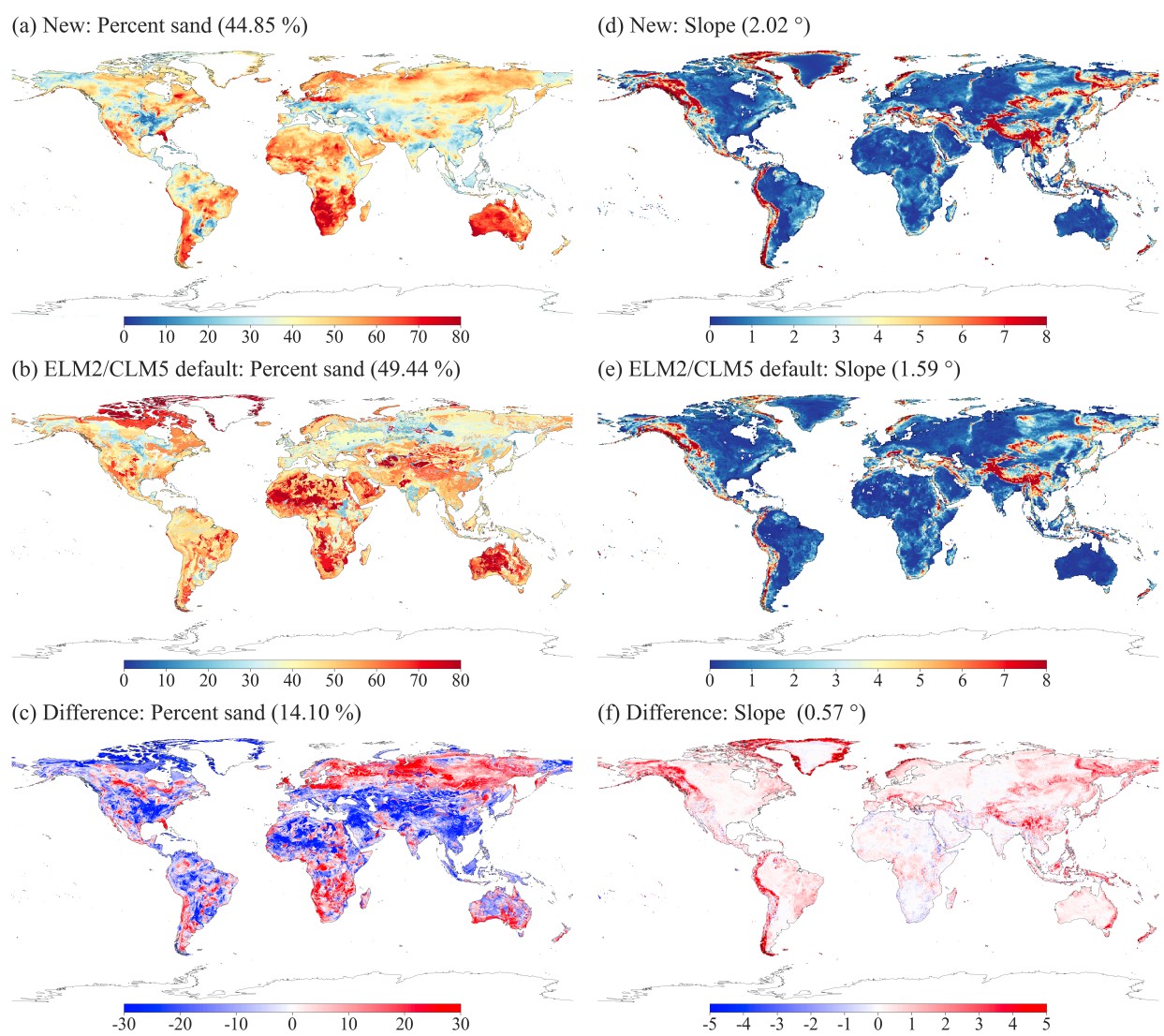

Figure 6. Comparisons of percent sand and slope. (a) new and (b) ELM2/CLM5 default percent sand, along with (c) their difference (new – ELM2/CLM5 default) for percent sand; (d) new, (e) ELM2/CLM5 default, and (f) their difference for slope. The global average is shown in the subplot titles, with the global average of the absolute difference provided for (c) and (f).

**4.3 Demonstration 1km simulation over CONUS**

ELM simulations at a 1 km resolution display significant spatial heterogeneity over CONUS (Figure 7). The values of SM, LH, ELR, and ASR across CONUS follow approximately normal

distributions, with averages of 0.3 m$^3$/m$^3$, 39.0 W/m$^2$, 371.7 W/m$^2$, 156.7 W/m$^2$, respectively (as
shown in the histogram plots in Figure 7). SM shows drier conditions over the West and Southwest
and wetter conditions over the Midwest, Corn Belt, Mississippi River basin, and Northeast (Figure
7a). LH shows high values over the central and southeast, and lower values over the west and
southwest (Figure 7b). The ELR generally shows higher values over regions with high surface
temperature in the south (Figure 7c). The ASR shows higher values over the southwestern regions
determined by incoming solar radiation and albedo (Figure 7d). Despite the high-resolution
heterogeneity shown at 1 km resolution, we can still see the spatial patterns distinguished at coarse
resolution, i.e., 0.5º × 0.5º. These coarser footprints are from the GSWP3 atmospheric forcing with
0.5º resolution. As concluded by Li et al. (2022), atmospheric forcing is one primary heterogeneity
source for land surface modeling. Therefore, k-scale atmospheric forcing needs to be developed to
further advance k-scale offline land surface modeling.

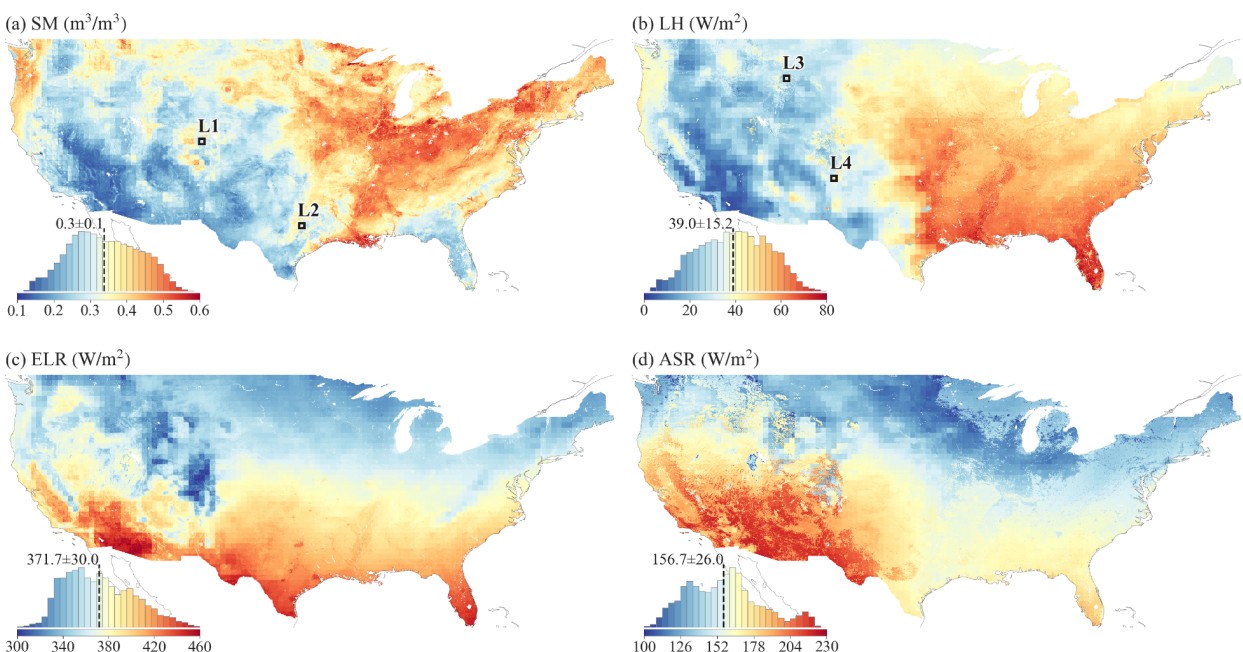


Figure 7. The annual mean of 1 km simulations of (a) SM, (b) LH, (c) ELR, and (d) ASR over
CONUS. The 0.5° × 0.5° boxes marked as L1, L2, L3, and L4 in (a) and (b) are selected to
demonstrate the spatial scaling analysis. The inserted histogram plot illustrates the distribution of
ELM2 simulations.

**4.4 Demonstration of spatial scaling across scales**
We next demonstrate the relationships between spatial variabilities and spatial scales for SM and
LH. Four locations (in Figures 4a and 4b) are specifically chosen to showcase varying levels of
spatial information loss: L1 and L3 demonstrate a relatively large loss for SM and LH, respectively,
while L2 and L4 represent a relatively small loss for SM and LH, respectively.
At location L1 (Figure 8a), when the 1 km simulation is upscaled to coarser resolutions (i.e., larger
spatial scale ratios), the spatial variability of SM decreases, resulting in a negative slope of $\beta$. As
shown in Figure 9a, compared to the original 1 km resolution, the information loss $\gamma$ reaches up to
54.9% at the 12 km spatial scale. The spatial pattern of SM is consistent with the spatial pattern of
percent clay (Figures 6a vs. 6b and 6c vs. 6d), indicating that soil texture contributes significantly
to the spatial variability of SM. However, SM has a more negative $\beta$ than the percent clay ($\beta$ = –
0.28 vs. –0.19 at L1, as shown in Figure 8a), suggesting that SM variability is amplified likely by
other processes that are also influenced by soil texture. In contrast to location L1, location L2
exhibits less negative $\beta$ values for both SM and percent clay, suggesting that their spatial
variabilities exhibit less scale dependence (Figures 5a, 6c, and 6d). Both SM and percent clay at
location L2 approximately maintain their spatial patterns of high values in the west and low values
in the east across spatial scales (Figures 6c and 6d).
For LH, there is a more negative $\beta$ value at location L3 than at location L4 ($\beta$ = –0.27 at L3 vs. –
0.08 at L4, as shown in Figure 8b), which indicates a larger decrease of spatial variability across
spatial scales and lower variability persistence at location L3 than location L4 (Figure 10). The
spatial pattern of LH is consistent with the spatial pattern of LAI (Figures 7a vs. 7b and 7c vs. 7d)
at different spatial scales, suggesting that vegetation plays a significant role in the spatial
variability of LH. Similar to comparison between SM and soil texture, LH has a more negative β
than LAI (Figure 8b).

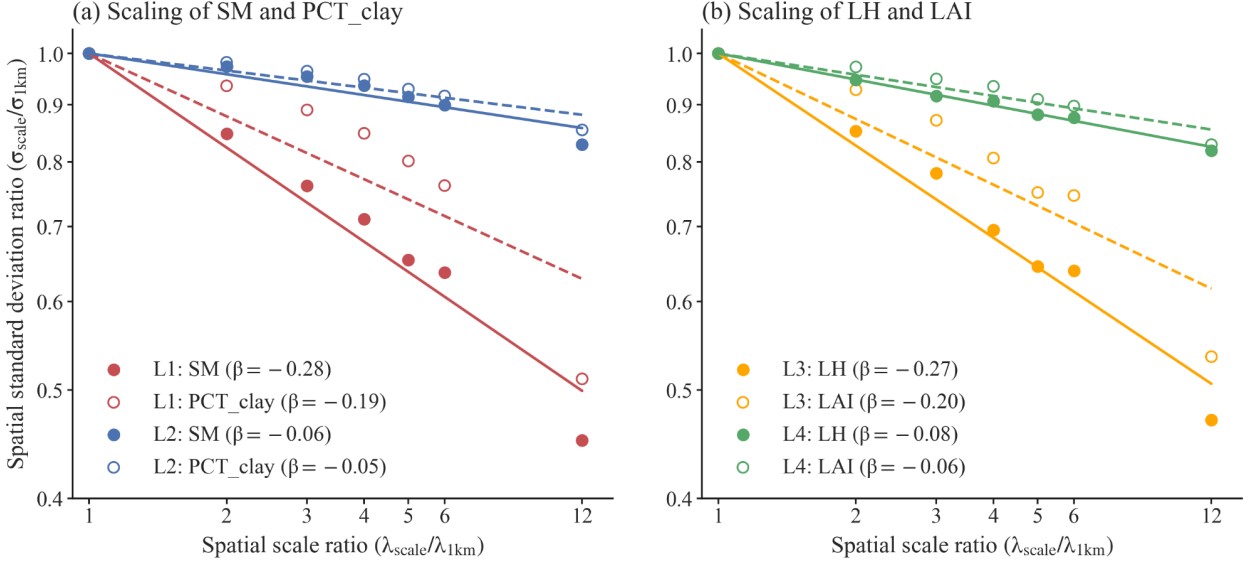


Figure 8. The scaling of spatial variabilities for (a) SM and percent clay, and (b) LH and LAI. Both
the x-axis and y-axis are in logarithmic scale. The slope of the linear regression line, β, quantifies
the strength of the negative relationship between spatial scale and spatial variability. A more
negative β value indicates a higher spatial-scale dependency and increased information loss at
coarser spatial scales. Four 0.5° × 0.5° boxes (displayed in Figure 7), namely L1 to L4, are chosen
to contrast larger and smaller negative β values for SM and percent clay (L1 and L2) and for LH
and LAI (L3 and L4).

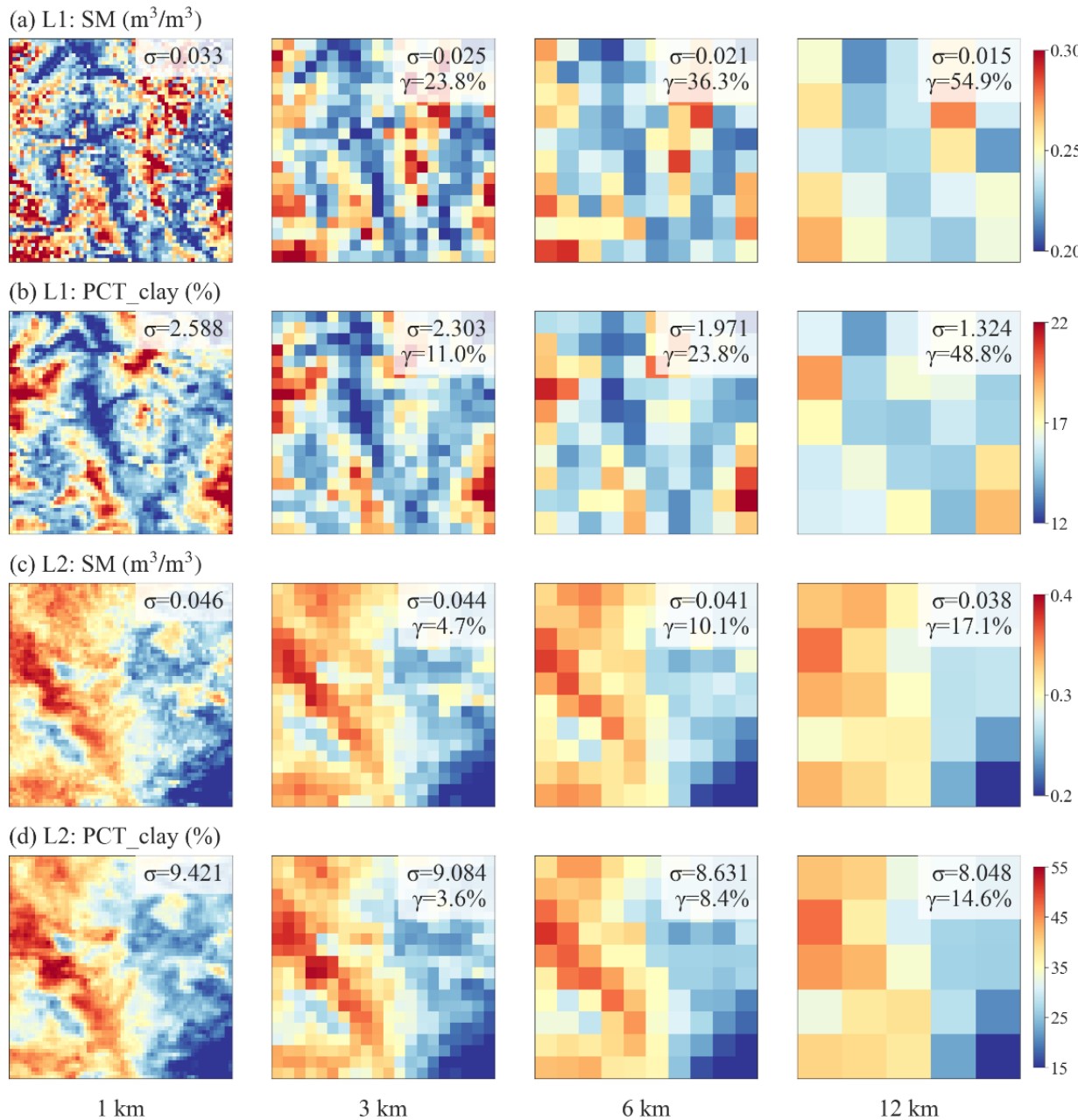


Figure 9. Comparison of SM and percent clay across spatial scales at locations L1 and L2 highlighted in Figure 7. Each subplot displays the spatial patterns of SM or percent clay within a 0.5° × 0.5° box, with the σ and γ presented in the legend.

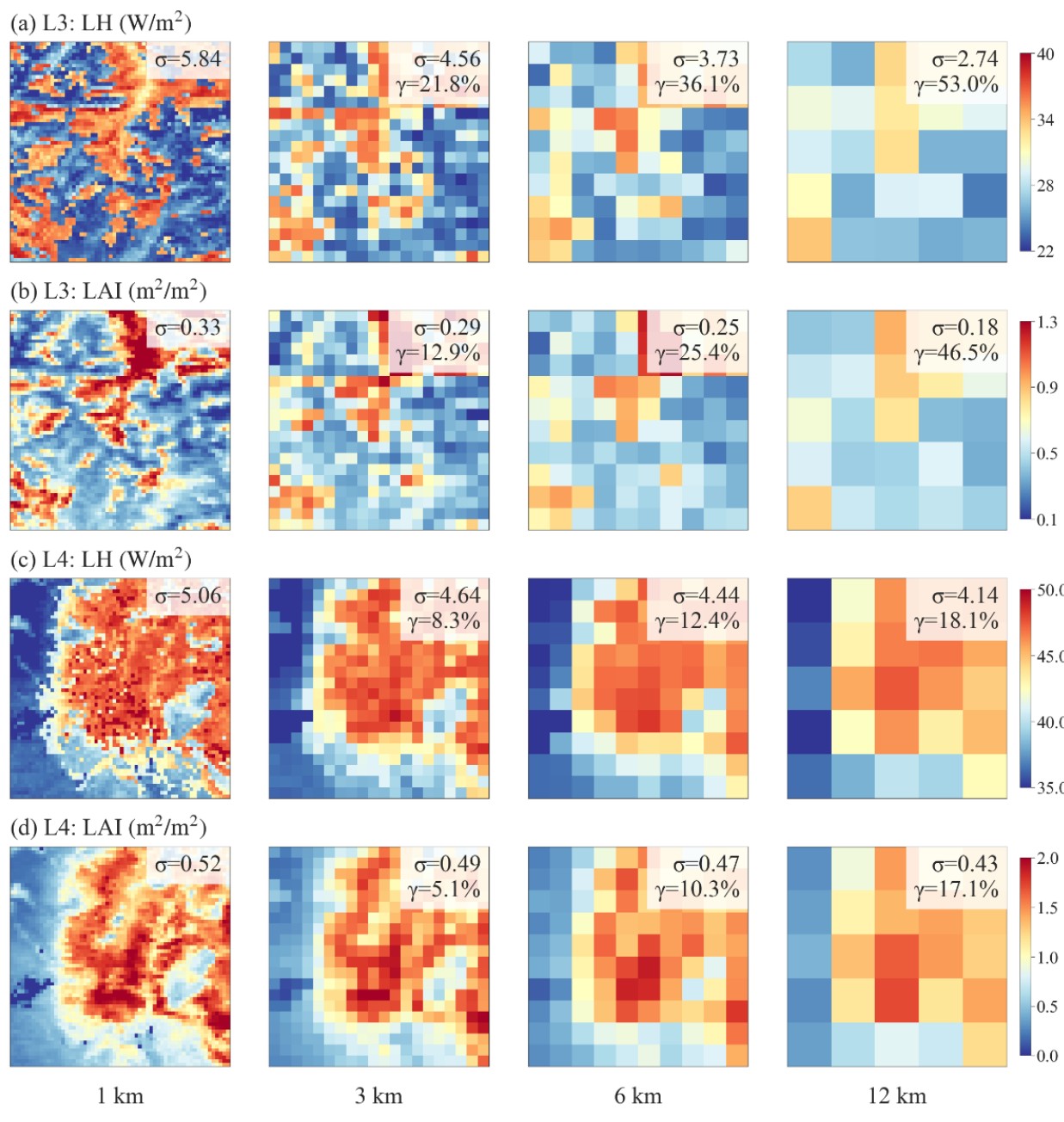


Figure 10. Similar to Figure 9, but for LH and LAI at locations L3 and L4.

**4.5 The spatial variability of water and energy simulations and their drivers**

We quantified the spatial variability simulated at 1 km resolution using σ within each 0.5º × 0.5º box across CONUS. Four ML models were built to explore the spatial relationships between σ and its potential drivers including σ of the land surface parameters and the temperature and precipitation averaged over the grid box. Overall, the ML models performed well in predicting the σ of the simulated variables, with small root mean square error (RMSE) and large $R^2$ (see Figure S24). SM shows larger spatial variability in the US Southern Coastal Plain, lower Mississippi River, Northeast, Southeast, and regions around the Great Lake (Figure 11a), which is roughly consistent with the spatial heterogeneity of the high-resolution SM simulation in Vergopolan et al. (2022). Based on the SHAP method, the spatial variability of SM across CONUS is driven by various factors, mainly including the spatial variabilities of percent sand and percent clay, mean precipitation, the σ and μ of soil organic matter, the σ of canopy height, and mean temperature (Figure 11b). Mean precipitation and temperature reflect climate conditions (Figure S26), which are related to the water supply and water demand of soil water content. The spatial heterogeneity of soil properties, such as texture and organic matter content, affects soil hydraulic properties and generate more spatially variable soil water content. Vegetation characteristics, such as canopy height and LAI, could influence SM spatial variability through their effect on roughness length and rooting depth.

The spatial variability of LH is large in the southeastern, central, and western mountainous regions of the US (Figure 11c). Vegetation properties and climate conditions mainly drive the variability of LH (Figure 11d). The μ and σ of LAI can affect transpiration and soil evaporation, while canopy height can influence surface roughness length and, in turn, evapotranspiration. Mean precipitation

and temperature reflect the overall climate conditions related to the water and energy available for
latent heat.
ELR and ASR exhibit large spatial variability mainly over the western US, with ASR additionally
showing significant spatial variability across the Northern US (Figures 8e and 8g). This variability
is primarily driven by climate conditions such as mean precipitation and temperature, topographic
features such as standard deviation of elevation and slope, and vegetation properties including LAI
and canopy height (Figures 8f and 8h). These factors are related to the radiation input and surface
properties, such as albedo and roughness length, which impact the energy cycles and availability
of ELR and ASR.

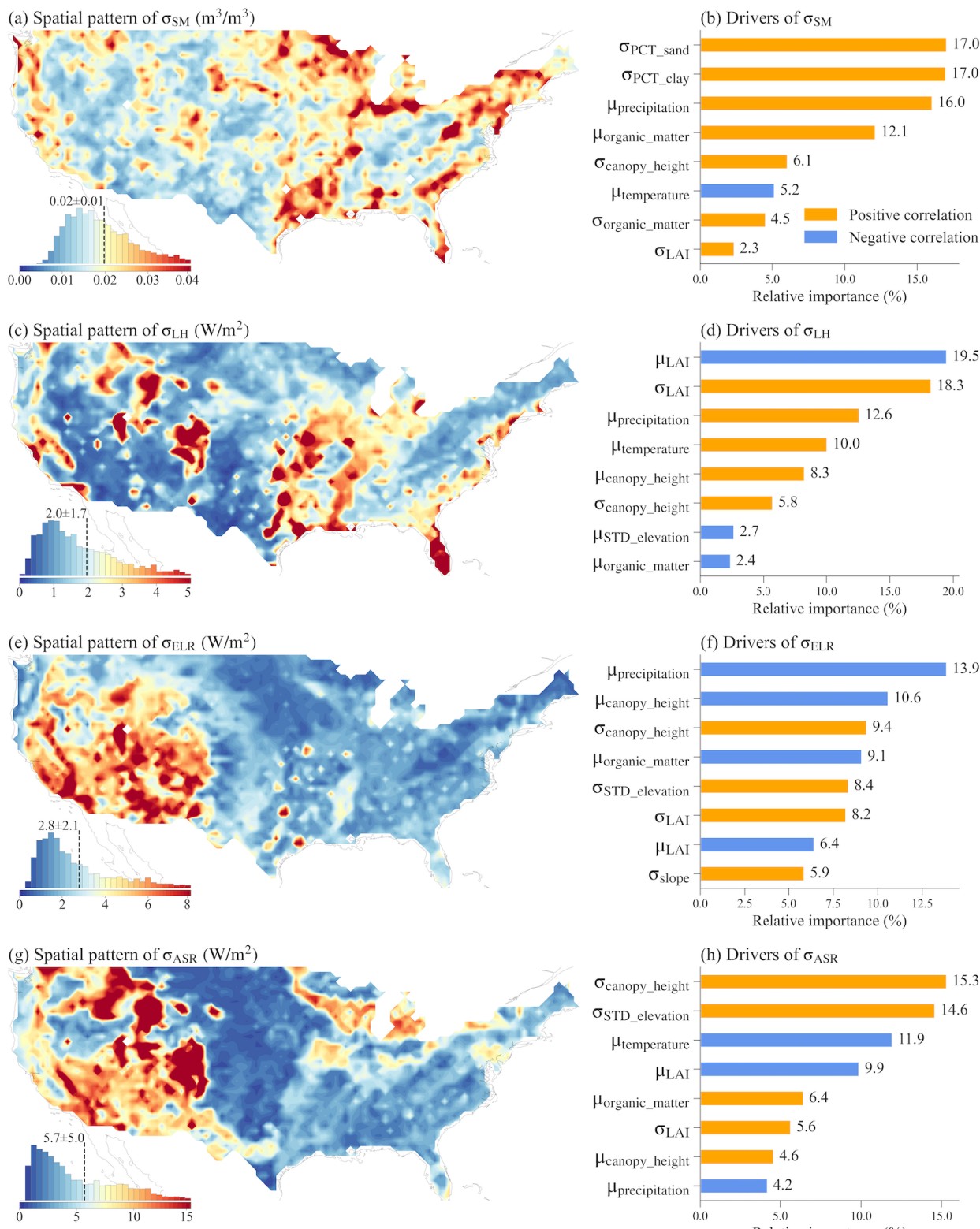


Figure 11. The spatial variability over each 0.5º × 0.5º grid cell (left plots) and the top eight most

important drivers (right plots) of the spatial variability for SM, LH, ELR, and ASR. The inserted

histogram plot illustrates the probability distribution of the spatial variability across CONUS. The
relative importance of each variable in determining the spatial variability is calculated as the ratio
of the mean |SHAP value| of the variable to the sum of the mean |SHAP value| of all variables.
Therefore, the sum of the relative importance of all variables is 100%.

**4.6 The information loss of water and energy simulations and their drivers**
We also evaluated the information loss in simulations when upscaling from 1 km to 12 km
resolution and analyzed the drivers of their spatial patterns over CONUS. Four ML models were
built to explore the relationships between the γ of the simulations and its drivers including the γ of
the land surface parameters and the mean temperature and precipitation averaged over the 0.5º ×
0.5º box. These ML models performed well in predicting the simulations' γ, with small RMSE and
large $R^2$ (Figure S25).
Significant information loss ranging from 31% to 54% with maximum values exceeding 90% is
observed for SM, LH, ELR, and ASR simulations (Figure 12). Their spatial patterns and drivers
show distinct variations. $\gamma_{SM}$ is primarily driven by the information loss of percent clay and sand,
mean soil organic matter, and mean temperature, which affects the soil hydraulic properties and
soil water balance (Figures 9a and 9b). $\gamma_{LH}$ displays high values in the eastern US and low values
in the western US (Figure 12c). It is primarily contributed by the information loss of vegetation
properties such as LAI and canopy height, and mean LAI, which influences the partitioning of LH
and sensible heat, and the partitioning of transpiration and evaporation (Figure 12d). $\gamma_{ELR}$ exhibits
high values in the central and eastern US, particularly in the northeastern US, while $\gamma_{ASR}$ has high
values almost all over the US, especially in the eastern regions (Figures 9e and 9g). $\gamma_{ELR}$ and $\gamma_{ASR}$
are largely driven by vegetation properties such as LAI and canopy height, which are associated
with energy processes such as albedo (Figures 9f and 9h). Additionally, topography factors of
standard deviation of elevation and slope also slightly contribute to $\gamma_{ASR}$.

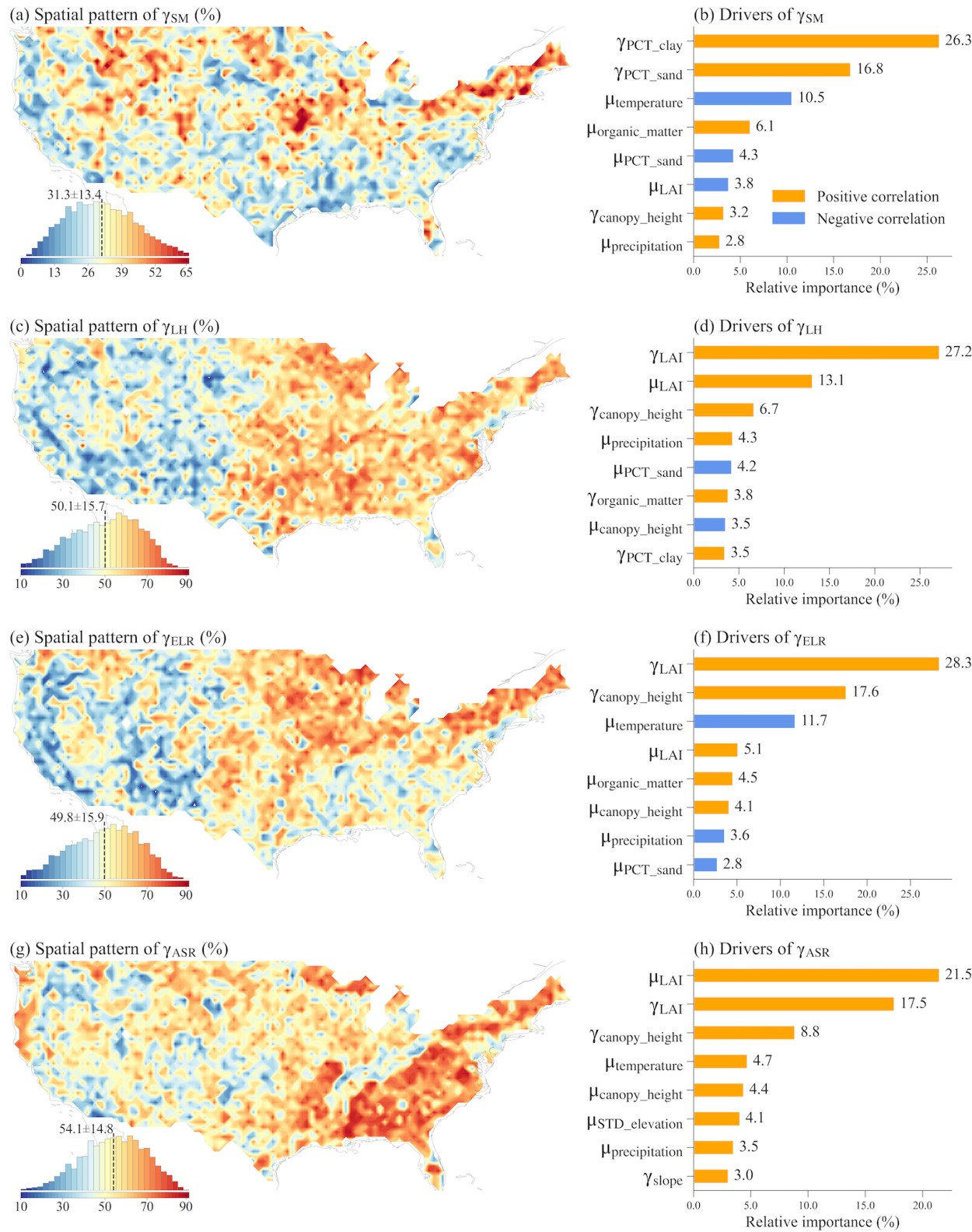


Figure 12. Same to Figure 11 but for information loss.

**4.7 Comparison of ELM simulation against reference data**

The average spatial biases between ELM and reference datasets across CONUS are relatively small, with SM bias at -0.01 m³/m³, LH bias at 1.8 W/m², ELR bias at -3.8 W/m², and ASR bias at 1.1 W/m² (Figure 13 and Figure S27). The correlation coefficient ($R^2$) between ELM and reference datasets was relatively high at 0.60 (for SM), 0.70 (for LH), 0.96 (for ELR), and 0.90 (for ASR). However, the spatial distribution of these biases exhibits variability, with some areas showing more pronounced biases than others. Specifically, in comparison with GLEAM SM, ELM tends to underestimate SM in the southeastern Texas and across the eastern and southeastern CONUS, while it overestimates SM in the western, central, and southwestern CONUS, including the central eastern US which are primarily agricultural areas. For LH, ELM simulates higher values than the MODIS LH dataset in the western and central US and Florida, but lower values in regions such as the eastern and northeastern CONUS, the western US coastal areas, and the Pacific Northwest. Regarding radiation variables, ELM generally underestimates ELR across nearly all of CONUS and tends to overestimate ASR, particularly in the southwestern, southern, eastern, northeastern, and northern regions of CONUS.

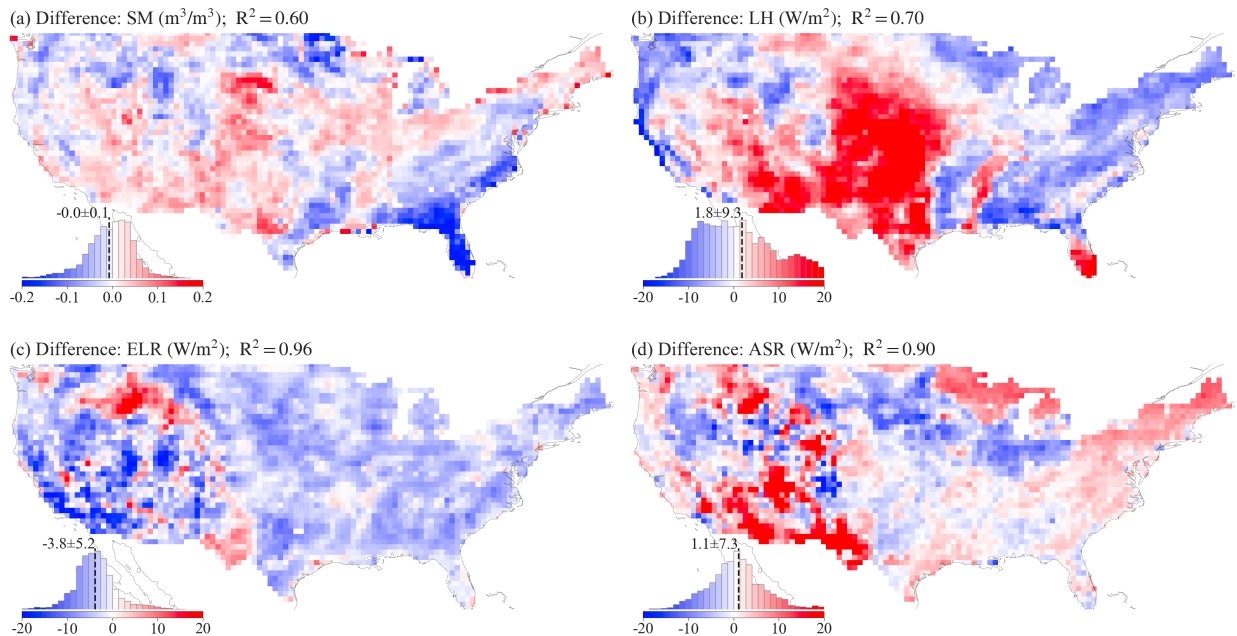


Figure 13. Annual mean bias between ELM-simulated variables and reference datasets over
CONUS: (a) SM, (b) LH, (c) ELR, and (d) ASR. The negative values indicate lower ELM values
compared to the reference data. The inserted histogram plot illustrates the distribution of grid
values. For spatial patterns of the reference datasets, refer to Figure S27. The correlation
coefficient (R²) between the ELM simulation and the reference dataset is calculated and displayed
in the title of each subplot."

## 5. Discussion

The development of new 1 km land surface parameter datasets in this study marks a substantial improvement over commonly used land surface parameters such as CLM5 and K2012, leveraging the latest high-resolution data sources with rigorous validation, including MODIS PFTs, enhanced LAI and canopy height, soil properties, and topography factors. When compared with K2012 and ELM2/CLM5 default datasets, the new 1k parameters exhibit notable differences, suggesting potential improvement due to the use of more advanced data sources. Distinct features of the new parameters include a reduction in bare soil compared to ELM2/CLM5, especially in regions like North America and Central Asia, and diverse coverage of specific PFTs such as NDT and BET-Tropical in areas like Siberia and South America. The LAI of the new parameters diverges from K2012 and ELM2/CLM5, showing lower values in NET-Boreal PFT of the northern hemisphere but higher BET-Tropical PFT in the tropics. The soil parameters, particularly in regions like Europe, Central Asia, and the Western US, show significant differences between the new and ELM2/CLM5 defaults. Moreover, the new parameters indicate larger slopes in mountainous regions and more distinct elevation differences in areas such as Greenland and the Tibetan Plateau compared to ELM2/CLM5. These differences potentially highlight enhanced accuracy and sophistication of the new 1k parameters. Their enhanced resolution and rigorous validation suggest a substantial capacity to improve ESMs modeling. Additionally, the richness of multi-year data for LULC, LAI, and SAI in these datasets is especially valuable for examining land use and cover changes, urbanization trends, deforestation impacts, and agricultural transformations.

The new 1 km land surface parameters can improve k-scale offline LSMs modeling by better capturing spatial surface heterogeneity. As evidenced by the 1 km ELM simulation over CONUS,

soil properties, vegetation properties, and topographic factors contribute a lot to the spatial heterogeneities of ELM water and energy simulations. Upscaling 1 km to a coarser 12 km resolution, we observe significant spatial information loss, with SM experiencing an average loss of 31%, and LH, ELR, and ASR experiencing around 50% information loss on average (Figure 12). This conclusion is in line with the results of Vergopolan et al. (2022), which showed a substantial loss of spatial information in soil moisture when upscaling from 30 m to 1 km resolution, with an average loss of approximately 48% and up to 80% over the CONUS region. The XML analysis reveals that the spatial variability and information loss of ELM2 simulations are influenced by the spatial variability and information loss of the different variables of land surface parameters, as well as the mean precipitation and temperature (Figures 11 and 12). Our findings highlight the critical role of land surface parameters in contributing to the spatial variability of water and energy in land surface simulations, showcasing the value of the developed high-resolution datasets. Another implementation example where our 1 km land surface parameters can be beneficial is in hillslope-scale simulations, which are fundamental for organizing water, energy, and biogeochemical processes (Fan et al., 2019). Krakauer et al. (2014) have highlighted the significance of between-cell groundwater flow, which becomes comparable in magnitude to recharge at grid spacings smaller than 10 km. Advancements have been made in ESMs to address hillslope-scale processes, including the representation of intra-hillslope lateral subsurface flow within grid cells in CLM5 (Swenson et al., 2019), the development of explicit lateral flow processes between grid cells (Qiu et al., 2023), and the incorporation of topographic radiation effects within and between grid cells (Hao et al., 2021). Another notable example is the integrated hydrology-land surface model ParFlow-CLM, which incorporates three-dimensional groundwater flow, two-dimensional overland flow, and land surface exchange processes (Maxwell, 2013).

ParFlow-CLM has demonstrated remarkable reliability in reproducing hydrologic processes, such
as its simulations at 3 km resolution for pan-European and 1 km resolution for CONUS (Naz et al.,
2023; O'Neill et al., 2021). More recently, Fang et al. (2022) coupled ParFlow with ELM and the
Functionally Assembled Terrestrial Ecosystem Simulator (FATES) to simulate carbon-hydrology
interactions at hillslope scale. By incorporating our 1 km datasets and leveraging these
advancements, we can improve simulations of hillslope-scale processes and enhance our
understanding of water and energy dynamics within ESMs.

Additionally, the new land surface parameters are also a timely resource for supporting the
emerging need for k-scale Earth system modeling, particularly in improving land-atmosphere
interaction processes. Representing the impact of spatial heterogeneity on land-atmosphere
interaction processes is a major challenge in Earth system modeling. Taking E3SM as an example,
researchers have proposed three key approaches to enhance spatial heterogeneity representation to
address this challenge. In line with these approaches, our newly developed 1 km land surface
parameters offer promising opportunities for improving land-atmosphere coupling within ESMs.
The first approach to enhance the representation of spatial heterogeneity is to directly conduct
simulations at high resolution. For instance, the Simple Cloud-Resolving E3SM Atmosphere
Model (SCREAM) has been used to perform global simulations at 3.25 km (Caldwell et al., 2021),
although the land surface parameters were based on coarser resolution datasets. By utilizing the
new 1 km land surface parameters, we can enhance the representation of land surface heterogeneity
within the ELM component of SCREAM, potentially improving modeling of land–atmosphere
coupling. The second and third approaches focus on improving the representation of land surface
heterogeneity within ESMs run at a coarse resolution while accounting for subgrid heterogeneity
in two different ways. In the second approach, the Cloud Layers Unified By Binormals (CLUBB)
has been implemented in E3SM Atmosphere Model (EAM) version 1 (Rasch et al., 2019;
Bogenschutz et al., 2013), to better account for subgrid atmospheric heterogeneity of turbulent
mixing, shallow convection, and cloud macrophysics. Recently, Huang et al. (2022) developed a
novel land-atmosphere coupling scheme in EAM that enables the communication of subgrid land
surface heterogeneity information to the atmosphere model with CLUBB, significantly impacting
boundary layer dynamics. The new 1km datasets can provide more accurate land surface
representations of the variability of individual patches and the inter-patch variability that were
used in Huang et al. (2022). The third approach is the Multiple Atmosphere Multiple Land (MAML)
approach used in the multiscale modeling framework (MMF) in which a cloud resolving model
(CRM) is embedded within each grid cell of the atmosphere (Baker et al., 2019; Lin et al., 2023;
Lee et al., 2023). In the MAML approach, each CRM column within the atmosphere grid is coupled
directly with its own independent land surface. This enables a more explicit representation of the
impact of spatial heterogeneity on land-atmosphere interactions within each grid and has shown
notable impacts on water and energy simulations (Baker et al., 2019; Lin et al., 2023). Lee et al.
(2023) highlighted the limitation of the current MAML approach, which utilizes the same land
surface characteristics for each land surface model interacting with the CRM column within the
same grid, which could lead to a weak representation of land-atmosphere interactions. To address
this limitation, incorporating the new 1 km land surface parameters within the MAML approach
can provide more detailed information about land surface heterogeneity, enabling a more accurate
capture of land-atmosphere interactions.

Evaluation of k-scale simulations, while essential, faces significant challenges as merely updating
the land surface input data to the new 1k parameters for k-scale simulations does not guarantee
improved model performance. This is clearly evidenced in our ELM demonstration simulations,
where, despite relatively low CONUS averaged biases for water and energy simulations, the spatial
variation in these biases cannot be overlooked, with some regions exhibiting notably larger biases.
It is important to emphasize that enhancing model performance requires not just updated input
data, but also appropriate calibration of model parameters and faithful model structures to
represent various processes. First, LSMs and ESMs that have been adapted for simulations at
coarser resolutions commensurate with the resolutions of previous land surface data require
recalibration for effective high-resolution modeling. This necessity for recalibration is echoed by
Ruiz-Vásquez et al., (2023), who noted that updating the ECMWF system with new land surface
data did not inherently improve performance, but improvements were seen after recalibrating key
soil and vegetation-related parameters. Second, high-resolution modeling requires the
incorporation of new physical processes crucial at finer scales. For example, hillslope-scale
processes like lateral flow and topography-radiation interactions are key to water and energy fluxes
at high resolution (Han et al., 2023; Hao et al., 2021). With increased heterogeneity at higher
resolutions, larger differences in land surface properties such as vegetation water use strategies
requires more attention to plant hydraulics besides the traditional focus on soil hydraulics for a
more accurate depiction of plant water use, as highlighted by Li et al., (2021). Third, the lack of
high-resolution benchmarks for large-scale applications, like k-scale atmospheric forcing data,
remains a challenge, despite the availability of relative coarse resolution global datasets such as
ERA5_Land (Muñoz-Sabater et al., 2021) and MSWX (Beck et al., 2021). Additionally, using soil
moisture as an example, multiple high-resolution datasets exhibit significantly different
performance when compared to in-situ measurements (Beck et al., 2021). Lastly, when evaluating
simulations against benchmarks, it is crucial not only to assess absolute differences using metrics
like bias and root mean square error but also to examine other metrics, such as the relationships
between physical variables (e.g., rainfall vs. runoff; soil moisture vs. evapotranspiration),
information loss, and the tail quantiles of the probability distribution functions for simulations (e.g.,
extreme events, Li et al., 2020).

There are certain opportunities for future development of 1k parameters. The urban extension may
vary based on data sources, urban definitions, and the algorithms employed, such as those derived
from harmonized nighttime lights (Zhao et al., 2022), global artificial impervious area (GAIA, Li
et al., 2020b; Gong et al., 2020), urban expansion (Liu et al., 2020; Kuang et al., 2021),
necessitating careful consideration in specific modeling applications. Additionally, urban
classification in J2010, based on global building height data, is limited by the lack of a consistent
and publicly accessible global dataset, despite available regional data for Europe (Frantz et al.,
2021), the US (Li et al., 2020a), and China (Cao and Huang, 2021; Yang and Zhao, 2022), thus
posing challenges to future urban classification enhancements. Incorporating local climate zones
offers a promising approach for urban classification and modeling. Moreover, the multiple-year
high-resolution PFT maps like the ones developed by the European Space Agency's Climate
Change Initiative could be used to further extend this dataset for a longer period (Harper et al.,
2023). Soil color, crucial for soil albedo and surface energy balance, lacks extensive global datasets
for ESMs modeling, but the global soil color map derived by Rizzo et al. (2023) offers potential
for further kilometer-scale ESMs and LSMs modeling.

The strategic aggregation of high-resolution parameters to coarser resolutions are crucial to
maintain accuracy and effectiveness in modeling applications. For instance, in soil properties, the
basic parameters (e.g., percent sand) are often utilized to derive secondary parameters (e.g.,
saturated water content). This aggregation procedure, whether performs before or after deriving
secondary parameters—known as 'aggregating first' and 'aggregating after'—is influenced by the
non-linear relationships between basic and derived parameters, with the latter method generally
preferred (Shangguan et al., 2014; Dai et al., 2019). Our study's initial approach in upscaling soil-
and topography-related parameters follows the 'aggregate first' approach, aligning with the
structure of models like ELM2 and CLM5. Conversely, models such as Common Land Model
(CoLM, Dai et al., 2003) and community Noah with multi-parameterization options (Noah-MP,
He et al., 2023; Niu et al., 2011; Yang et al., 2011) integrate secondary derived soil related
parameters directly as inputs, effectively demonstrating the advantages of the 'aggregating after'
approach. By leveraging secondary derived parameters from comprehensive databases such as
SoilGrids (Hengl et al., 2017) and GSDE (Shangguan et al., 2014), these models provide a valuable
framework for future development of models like ELM2 and CLM5 by directly integrating
secondary derived parameters.

**6. Data availability**

The 1 km land surface parameters are publicly available at Zenodo: https://zenodo.org/records/ 10815170 (Li et al., 2024) and PNNL Datahub: https://doi.org/10.25584/PNNLDH/1986308 (Li et al., 2023).

**7. Conclusions**

We developed 1 km global land surface parameters using the latest available datasets covering multiple years from 2001 to 2020. These parameters comprise four categories: LULC of PFTs and non-vegetative land cover, vegetation properties, soil properties, and topographic factors. The new 1k parameters, when compared to the K2012 and ELM2/CLM5 default datasets, display significant differences, indicating their potential superiority stemming from the utilization of latest and more advanced data sources. The 1 km resolution ELM simulations conducted over CONUS demonstrate the valuable capabilities of the new datasets in enabling k-scale land surface modeling. Through scaling analysis of the 1 km resolution simulations within 0.5º × 0.5º boxes where spatial heterogeneity of the simulations is induced only by spatial heterogeneity of the land surface parameters, we revealed the significant impact of land surface parameters on the spatial variability of water and energy simulations. The spatial information loss of these simulations over CONUS is significant when upscaling from 1 km to a coarser 12 km resolution, with an average ranging from 31% to 54% and up to more than 90%. The XML analysis reveals that the spatial variability and spatial information loss of ELM2 simulations are primarily impacted by the spatial variability and information loss of soil properties, vegetation properties and topography factors, as well as the mean climate conditions of precipitation and temperature. Furthermore, the spatial variability of water and energy in the 1 km simulations is not dominated by the spatial heterogeneity of any land

surface parameters, suggesting the usefulness of the multi-parameter high-resolution land surface
parameter dataset. Furthermore, the comparison against four benchmark datasets indicates that
ELM generally performs well in simulating soil moisture and surface energy fluxes. The
availability of 1 km land surface parameters is a valuable resource that addresses the emerging
needs of k-scale LSMs and ESMs modeling. By providing accurate and precise information, these
1 km land surface parameters will significantly enhance our understanding of the water, carbon,
and energy cycles under global change.

**Author contributions**

LL, GB, and DH designed the study, processed the datasets, conducted experiments, and drafted the manuscript. LRL contributed to the conceptual design, discussion of results, and manuscript revisions.

**Acknowledgments**

This study is supported by the US Department of Energy (DOE) Office of Science Biological and Environmental Research as part of the Regional and Global Model Analysis (RGMA) program area through the collaborative, multi-program Integrated Coastal Modeling (ICoM) project. This study used DOE's Biological and Environmental Research Earth System Modeling program's Compy computing cluster at Pacific Northwest National Laboratory. Pacific Northwest National Laboratory is operated for the US Department of Energy by Battelle Memorial Institute under contract DE-AC05-76RL01830. DH acknowledges the support from the US DOE, Office of Science, Office of Biological and Environmental Research, Earth System Model Development program area, as part of the Climate Process Team projects. Our thanks to Ye Liu and Teklu Tesfa at PNNL for guidance on the canopy height dataset and K2012 datasets, respectively. We deeply appreciate the reviewers for their valuable insights and suggestions.

**Financial support**

This work was supported by the Regional and Global Modeling and Analysis program area of the US Department of Energy, Office of Science, Office of Biological and Environmental Research, as part of the multi-program, collaborative integrated Coastal Modeling (ICoM) project (grant no. KP1703110/75415).


**Competing interests**

At least one of the (co-)authors is a member of the editorial board of the Earth System Science

Data. The authors have no other competing interests to declare.

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

Index products for land surface and climate modelling, Remote Sens Environ, 115, 1171–1187,
https://doi.org/10.1016/j.rse.2011.01.001, 2011.
Yuan, X., Ji, P., Wang, L., Liang, X., Yang, K., Ye, A., Su, Z., and Wen, J.: High-Resolution
Land Surface Modeling of Hydrological Changes Over the Sanjiangyuan Region in the Eastern
Tibetan Plateau: 1. Model Development and Evaluation, J Adv Model Earth Sy, 10, 2806–2828,
https://doi.org/10.1029/2018ms001412, 2018.
Zeng, X., Shaikh, M., Dai, Y., Dickinson, R. E., and Myneni, R.: Coupling of the Common Land
Model to the NCAR Community Climate Model, J Climate, 15, 1832–1854,
https://doi.org/10.1175/1520-0442(2002)015<;1832:cotclm>2.0.co;2, 2002.
Zhao, M., Cheng, C., Zhou, Y., Li, X., Shen, S., and Song, C.: A global dataset of annual urban
extents (1992–2020) from harmonized nighttime lights, Earth Syst Sci Data, 14, 517–534,
https://doi.org/10.5194/essd-14-517-2022, 2022.
Zhou, Y., Li, D., and Li, X.: The Effects of Surface Heterogeneity Scale on the Flux Imbalance
under Free Convection, J Geophys Res Atmospheres, 124, 8424–8448,
https://doi.org/10.1029/2018jd029550, 2019.