# Peer review of "1Global 1km Land Surface Parameters for Kilometer-Scale Earth System Modeling"

_Earth System Science Data, 2023_

## Author Comment (AC1)

**Responses to comments <essd-2023-242>**

Dear Editor and referees,

We greatly appreciate your insightful comments and suggestions, and your time in reviewing our manuscript, which have helped us improve our manuscript. In response to your comments, we have made several significant changes in the revised manuscript, which are summarized below:

- In response to Reviewer 1's feedback, we have enriched the manuscript with more detailed descriptions of the validation of each data source used in our dataset development. Additionally, we have conducted a comprehensive comparison between our newly developed 1k parameters and the K2012 and ELM2/CLM5 default parameters, with corresponding updates made to the methods, results, discussion, and conclusions sections.

- In response to Reviewer 2's feedback, we have added two new paragraphs in the discussion section 4. These additions focus on the challenges and considerations related to parameter aggregation and the evaluation of k-scale simulations, providing a more thorough exploration of these critical aspects.

The point-by-point responses to the specific comments are provided below in blue. All line numbers listed below correspond to those in the clean version of the revised manuscript. We hope that our modifications have addressed all the concerns raised, and we appreciate your consideration of our revised manuscript.

Sincerely,

Lingcheng Li and co-authors

*Referee 1*

This research presents a new set of global land surface parameters with a resolution of 1 km for multiple years from 2003 to 2020. This manuscript is well-structured. The figures are well produced. The English is good. The results are clearly presented. However, major issues should be addressed before this manuscript may be reconsidered for publication in the esteemed ESSD.

Thank you for your encouraging comments and suggestions for improving our manuscript. We provide a response for each comment as detailed below.

For a majority of data description papers in ESSD, a solid verification of the new data based on ground truth data and the comparison between the new data set and the existing mainstreaming data set are necessary and always included. Without such information, readers cannot fully understand whether the new data set is reliable and how much this data set has been improved compared with existing data sets. Consequently, the significance of this research cannot be highlighted. Therefore, I strongly recommend the authors to present a quantitative comparison between the new data set and mainstream data sets (e.g. CLM5 and K2012 datasets) based on already existing reference data or manually collected reference data.

Thank you for your suggestions regarding the two aspects of quantitative comparison: first with existing reference and benchmark data, and second with mainstream datasets such as K2012 and ELM2/CLM5 default.

1. On the first point, it is important to note that the 1k parameters are derived from datasets that have been rigorously validated and described in the literature. Consequently, we have chosen not to duplicate the evaluation of these source datasets. Instead, we have expanded our manuscript to include details about the validation undertaken for each data source.

- LULC parameters in L147–155.
  *In this study, the MODIS MCD12Q1 version 6 (Friedl et al., 2022) was employed to ascertain the Plant Functional Types (PFT) as well as other non-vegetative land categories at a spatial resolution of 1 km spanning the years 2001 to 2020. The integrity of the MODIS land cover product has been established through a 10-fold cross-validation accuracy assessment using the Terrestrial Ecosystem Parameterization database (Sulla-Menashe et al., 2019). This land*

*cover product offers richer and more flexible land cover data with higher accuracy and substantially less year-to-year stochastic variation in classification results (Sulla-Menashe et al., 2019). Being the sole operational global land cover product available with annual intervals, it addresses a significant gap in the realm of global change research.*

- LAI parameters in L196–201.

  *BNU_LAI, an enhanced version of the MODIS LAI product, has been subjected to thorough quality control, incorporating multiple algorithms for improved accuracy (Yuan et al., 2011). Its validation involved an extensive array of LAI reference maps and employed the bottom-up approach advocated by the CEOS Land Product Validation sub-group (Morisette et al., 2006). Compared to the original MODIS LAI, the BNU_LAI dataset exhibits superior performance, along with enhanced spatiotemporal continuity and consistency.*

- Canopy height parameters in L210–L217.

  *We leveraged a global vegetation canopy height dataset sourced from Lang et al. (2023). This dataset, derived using a probabilistic deep learning model, fuses Sentinel-2 images with the Global Ecosystem Dynamics Investigation (GEDI) to retrieve canopy height. It stands out as the inaugural global canopy height dataset offering consistent, wall-to-wall coverage at a 10 m spatial resolution across all vegetation types. Assessments using hold-out GEDI reference data and comparisons with independent airborne LiDAR data demonstrate that the approach outlined by Lang et al. (2023) produces a meticulously quality-controlled, state-of-the-art global map product, accompanied by quantitative uncertainty estimates.*

- Soil-related parameters in L224–227.

  *The soil product underwent rigorous quantitative evaluation using a cross-validation method, which ensures alignment with established pedo-landscape features and provides spatial uncertainty to guide product users (Poggio et al., 2021).*

- Topography-related parameters in L239–244.

  *We employed the digital elevation from the Multi-Error-Removed Improved-Terrain DEM (MERIT DEM, Yamazaki et al., 2019) to obtain topography-related parameters. The MERIT*

*DEM provides globally consistent elevation data at 90 m resolution, distinguished by its exceptional vertical accuracy. This accuracy was rigorously validated against ICESat's lowest elevations in both forested and non-forested regions and was further benchmarked using the UK's premium airborne LiDAR DEM (Yamazaki et al., 2019).*

2. Regarding the comparison with existing mainstream datasets, we have included a comprehensive comparison between our new 1k parameters and the K2012 and ELM2/CLM5 default parameters. This comparison has been elaborated in the methods, results, discussion, and supplementary sections of our manuscript. Considering that K2012 encompasses only LULC and LAI/SAI parameters, we focused our comparison on these aspects, contrasting LULC and LAI among the new 1k, CLM5, and K2012 datasets. For other parameters, including those related to soil and topography, we conducted comparisons between the new 1k and ELM2/CLM5 default parameters. We excluded the comparison of SAI, owing to limited data availability in K2012, and vegetation canopy height (top and bottom parameters) because the ELM2/CLM5 default parameters from the CESM input data repository only provide tabulated values for each PFT. Below are the detailed modifications included.

[revised manuscript text omitted]

- In the supplementary, corresponding Figures S1 to S23 have been added.

- In L566–584, the first paragraph of the discussion section has been updated.

*The development of new 1 km land surface parameter datasets in this study marks a substantial improvement over commonly used land surface parameters such as CLM5 and K2012, leveraging the latest high-resolution data sources with rigorous validation, including MODIS PFTs, enhanced LAI and canopy height, soil properties, and topography factors. When compared with K2012 and ELM2/CLM5 default datasets, the new 1k parameters exhibit notable differences, suggesting potential improvement due to the use of more advanced data sources. Distinct features of the new parameters include a reduction in bare soil compared to ELM2/CLM5, especially in regions like North America and Central Asia, and diverse coverage of specific PFTs such as NDT and BET-Tropical in areas like Siberia and South America. The LAI of the new parameters diverges from K2012 and ELM2/CLM5, showing lower values in NET-Boreal PFTs of the northern hemisphere but higher BET-Tropical PFTs in the tropics. The soil parameters, particularly in regions like Europe, Central Asia, and the Western US, show significant differences between the new and ELM2/CLM5 defaults. Moreover, the new parameters indicate larger slopes in mountainous regions and more distinct elevation differences in areas such as Greenland and the Tibetan Plateau compared to ELM2/CLM5. These differences potentially highlight enhanced accuracy and sophistication of the new 1k parameters. Their enhanced resolution and rigorous validation suggest a substantial capacity to improve ESMs modeling. Additionally, the richness of multi-year data for LULC, LAI, and SAI in these datasets is especially valuable for examining land use and cover changes, urbanization trends, deforestation impacts, and agricultural transformations.*

The remaining portions of the initial first paragraph in the discussion have been relocated and revised as the fifth paragraph, now found in lines 681–695.

*There are certain opportunities for future development of 1k parameters. The urban extension may vary based on data sources, urban definitions, and the algorithms employed, such as those derived from harmonized nighttime lights (Zhao et al., 2022), global artificial impervious area (GAIA, Li et al., 2020b; Gong et al., 2020), urban expansion (Liu et al., 2020; Kuang et al., 2021), necessitating careful consideration in specific modeling applications. Additionally, urban classification in J2010, based on global building height data, is limited by the lack of a consistent and publicly accessible global dataset, despite available regional data for Europe (Frantz et al.,*

*2021), the US (Li et al., 2020a), and China (Cao and Huang, 2021; Yang and Zhao, 2022), thus posing challenges to future urban classification enhancements. Incorporating local climate zones offers a promising approach for urban classification and modeling. Moreover, the multiple-year high-resolution PFT maps like the ones developed by the European Space Agency's Climate Change Initiative could be used to further extend this dataset for a longer period (Harper et al., 2023). Soil color, crucial for soil albedo and surface energy balance, lacks extensive global datasets for ESMs modeling, but the global soil color map derived by Rizzo et al. (2023) offers potential for further kilometer-scale ESMs and LSMs modeling.*

Another important issues is about the citation in the text. The citation should be thoroughly revised. For instance, the list of more than 10 references in a line can provide readers no accurate information and a clear relation between the reference and the mentioned information. e.g. L49-L50 ... and biogeochemical cycles, as well as land and atmosphere coupling (Giorgi and Avissar, 1997; Chaney et al., 2018; Zhou et al., 2019; Liu et al., 2017; Bou-Zeid et al., 2020; Chen et al., 2020; Nitta et al., 2020; Vrese et al., 2016)…These references should be clearly cited and explained. Personally, I do not suggest a list of more than 3 references in a line. The citations in the text are poor and all citations throughout the manuscript should be double-checked and revised to the right form.

Thank you for your valuable feedback and suggestions on improving the manuscript. We have meticulously updated the references as advised and conducted a thorough review of the manuscript to ensure accuracy of all citations.

Specifically, in L50–L53:

*High-resolution modeling can better capture the land surface heterogeneity and could improve simulations of terrestrial water and energy cycles (Giorgi and Avissar, 1997; Chaney et al., 2018; Xu et al., 2023), biogeochemical cycles (Chaney et al., 2018), as well as land–atmosphere coupling (Liu et al., 2017; Zhou et al., 2019; Bou-Zeid et al., 2020).*

In L685–L689:

*Additionally, urban classification in J2010, based on global building height data, is limited by the lack of a consistent and publicly accessible global dataset, despite available regional data for Europe (Frantz et al., 2021), the US (Li et al., 2020a), and China (Cao and Huang, 2021; Yang and Zhao, 2022), thus posing challenges to future urban classification enhancements.*

---

## Author Comment (AC2)

**Responses to comments <essd-2023-242>**

Dear Editor and referees,

We greatly appreciate your insightful comments and suggestions, and your time in reviewing our manuscript, which have helped us improve our manuscript. In response to your comments, we have made several significant changes in the revised manuscript, which are summarized below:

- In response to Reviewer 1's feedback, we have enriched the manuscript with more detailed descriptions of the validation of each data source used in our dataset development. Additionally, we have conducted a comprehensive comparison between our newly developed 1k parameters and the K2012 and ELM2/CLM5 default parameters, with corresponding updates made to the methods, results, discussion, and conclusions sections.

- In response to Reviewer 2's feedback, we have added two new paragraphs in the discussion section 4. These additions focus on the challenges and considerations related to parameter aggregation and the evaluation of k-scale simulations, providing a more thorough exploration of these critical aspects.

The point-by-point responses to the specific comments are provided below in blue. All line numbers listed below correspond to those in the clean version of the revised manuscript. We hope that our modifications have addressed all the concerns raised, and we appreciate your consideration of our revised manuscript.

Sincerely,

Lingcheng Li and co-authors

*Referee 2*

This manuscript highlights the importance of developing new global land surface parameters with a resolution of 1 km for Earth system models (ESMs) running at the kilometer scale. The study demonstrates that these parameters significantly impact the spatial heterogeneity and information loss in ESM simulations, particularly in relation to soil moisture, latent heat, emitted longwave radiation, and absorbed shortwave radiation. The use of eXplainable Machine Learning methods helps identify the influential factors driving this variability and information loss. The new land surface parameters have implications for advancing our understanding of water, carbon, and energy cycles under global change. The paper is well written and has significant value for high resolution LSM modeling. However, I have several concerns as shown in the following comments for the authors to be considered.

Thank you for your encouraging comments and suggestions for improving our manuscript. We provide a response for each comment as detailed below.

Major comments:

The aggregation order problem should be addressed when upscaling the secondary derived parameters including DEM-derived variables, PTF-dervived (Pedotransfer functions) soil parameters such as saturated water content. Previous studies have proved that this order has significant effect on the derived parameters and thus the modeling results. For example, the aggregation after method has been recommended by Shangguan et al. (2014) and (Dai et al., 2019). That is, you should first calculate the derived parameters at the high resolution and then aggregate them into low resolution. This is majorly due to the nonlinear relationship between the original parameters and derived ones.

However, the authors chose the aggregate first way according to the description in line 219~222, which is not a good way. At least, you should compare these two methods yourself, evaluate them and choose the better way.

Thank you for your insightful comments and recommendations regarding the aggregation order of secondary derived parameters, particularly in the context of soil parameters derived from DEM and PFTs. The studies by Shangguan et al. (2014) and Dai et al. (2019) effectively underscore the importance of the correct aggregation approach. For instance, in soil properties, the basic parameters (e.g., percent sand) are often utilized to derive secondary parameters (e.g., saturated water content). This aggregation procedure, whether performs before or after deriving secondary parameters—known as 'aggregating first' and 'aggregating after'—is influenced by the non-linear relationships between basic and derived parameters, with the latter method generally preferred (Shangguan et al., 2014; Dai et al., 2019).

In our initial approach when upscaling soil and topography-related parameters, we adopted an 'aggregate first' methodology, such as for the parameters related to soil and topography. We acknowledge that this method might not optimally preserve the accuracy of derived parameters. This choice was primarily influenced by the structure of current models like ELM2 and CLM5, where first-order parameters are inputs, and secondary derived parameters are computed within the model, precluding the 'aggregating after' approach for developing secondary derived parameters. We recognize this might not be the ideal method and suggest that future developments of these models should consider incorporating secondary parameters directly as inputs to mitigate potential inaccuracies.

In contrast, models such as CoLM (Dai, et al., 2014) and Noah-MP (He et al., 2023; Niu et al., 2011; Yang et al., 2011) use secondary derived parameters directly as inputs, facilitating the 'aggregating after' method. These models can utilize secondary derived parameters, like saturated water content, sourced from databases such as the Global Soil Dataset for Earth System Models (GSDE, Shangguan et al., 2014) and SoilGrids (Hengl et al., 2017). This approach highlights a crucial direction for future model development, advocating a shift towards using secondary derived parameters as direct inputs to enhance modeling accuracy and reliability.

Your feedback has been invaluable in highlighting this significant aspect of modeling methodology, and we will certainly consider this in our future research endeavors.

We added relevant content in the discussion:

In L697–712.

*The strategic aggregation of high-resolution parameters to coarser resolutions are crucial to maintain accuracy and effectiveness in modeling applications. For instance, in soil properties, the basic parameters (e.g., percent sand) are often utilized to derive secondary parameters (e.g., saturated water content). This aggregation procedure, whether performs before or after deriving secondary parameters—known as 'aggregating first' and 'aggregating after'—is influenced by the non-linear relationships between basic and derived parameters, with the latter method generally preferred (Shangguan et al., 2014; Dai et al., 2019). Our study's initial approach in upscaling soil-related parameters follows the 'aggregate first' approach, aligning with the structure of models like ELM2 and CLM5. Conversely, models such as Common Land Model (CoLM, Dai et al., 2003) and community Noah with multi-parameterization options (Noah-MP, He et al., 2023; Niu et al., 2011; Yang et al., 2011) integrate secondary derived soil related parameters directly as inputs, effectively demonstrating the advantages of the 'aggregating after' approach. By leveraging secondary derived parameters from comprehensive databases such as SoilGrids (Hengl et al., 2017) and GSDE (Shangguan et al., 2014), these models provide a valuable framework for future development of models like ELM2 and CLM5.*

[revised manuscript text omitted]

Minor comments:

Table 1: the LAI data is updated for 2000-2021, see the link: http://globalchange.bnu.edu.cn/research/laiv061

Thank you for highlighting this. We worked on the 1k datasets and manuscript before the extended LAI source data was published. Consequently, the current 1k data version doesn't incorporate these extended datasets, but we plan to include them in a future version.

Line 255: $\gamma$? You should explain it here.

We have revised the text in L312–L315.

*A more negative β indicates a larger dependency of data spatial variability on spatial scales, resulting in a higher information loss, denoted as $\gamma_{scale} = (1 - \sigma_{scale}/\sigma_{1\ km}) \times 100\%$. In this study, we focus on information loss at a 12 km scale, denoted as $\gamma_{12\ km}$. For simplicity in subsequent discussion, $\gamma_{12\ km}$ will be referred to as γ in the results section.*

Figure 4: why choose these four locations as they are located in the arid and semi-sarid regions? It is recommended to choose locations with a representation of various climate zones or PFT.

The four locations we selected serve solely for demonstration in the spatial scaling analysis. As the complete spatial scaling analysis for each CONUS grid, encompassing spatial variability and information loss, is depicted in Figures 8 and 9, we refrain from displaying additional grid demonstrations across different climate zones.

Line 340~341: the effect of soil texture is very likely amplified by the PTF-derived soil parameters especially soil hydraulic conductivity. Try to investigate this issue. In addition, try to answer questions like is the value of beta related to the clay content itself or other factors like soil heterology?

We acknowledge the potential impact of using Pedotransfer Function (PTF)-derived soil parameters on ELM2 simulations, which may have a more pronounced effect than soil texture. Due to the structural design of the model, soil texture serves as an input and PTF-derived parameters such as soil hydraulic conductivity are calculated internally within ELM2, and thus we have not performed analysis using PTF-derived soil hydraulic parameters. However, we emphasize the importance of considering secondary derived parameters as direct inputs, rather than relying solely on soil texture, for future model developments. This aspect is elaborated in the last paragraph of the discussion section, specifically at L688.

Regarding the slope of the linear regression line, β, it serves as a measure of the strength of the negative relationship between spatial scale and spatial variability. A more negative β value

indicates higher spatial-scale dependency, leading to increased information loss at coarser spatial scales. Thus, β is utilized to quantify the extent of standard deviation reduction across different scales. For a detailed explanation and visual representation, please refer to Figure 5 and its accompanying caption.

Figure 6: it shows that L1 has a lower standard deviation of clay, but a more negative beta than L2. This indicate that lower soil heterology does not lead to lower spatial-scale dependence. So, how to explain this? Also, the information loss is higher when the standard deviation is lower. How can it be? Same thing happened in Figure 7.

The definition of information loss is designed to quantify the relative changes in standard deviation as scaling from high to coarse resolution (see L316). Therefore, the information loss is associated with the rate of change in standard deviation across scales, rather than being directly tied to the absolute value of the standard deviation itself.

Figure 8 and 9: what is the link between these two figures? You may discuss this.

Figures 11 and 12 (previously Figures 8 and 9) are employed to highlight the impact of using the new 1k data on the spatial heterogeneity observed in water and energy simulations over CONUS. Figure 11 illustrates the spatial heterogeneity (i.e., the standard deviation) within each 0.5-degree grid cell, whereas Figure 12 demonstrates how this spatial heterogeneity diminishes when the 1 km simulations are upscaled to a coarser 12 km resolution (i.e., information loss). Essentially, the information loss depicted in Figure 12 serves as a secondary metric derived from the standard deviation presented in Figure 11. Figures 11 and 12 also elucidate the primary drivers influencing both the standard deviation and the information loss in our simulations.

The related discussion is elaborated in the second paragraph of the discussion section (L589–L597), where we have included specific references to these figures for enhanced clarity.

*Upscaling 1 km to a coarser 12 km resolution, we observe significant spatial information loss, with SM experiencing an average loss of 31%, and LH, ELR, and ASR experiencing around 50% information loss on average (Figure 12). This conclusion is in line with the results of Vergopolan et al. (2022), which showed a substantial loss of spatial information in soil moisture when upscaling from 30 m to 1 km resolution, with an average loss of approximately 48% and up to 80% over the CONUS region. The XML analysis reveals that the spatial variability and information loss of ELM2 simulations are influenced by the spatial variability and information loss of the different variables of land surface parameters, as well as the mean precipitation and temperature (Figures 11 and 12).*

---

## Author Response (AR2)

**Responses to comments <essd-2023-242>**

Dear Editor and referees,

We appreciate the opportunity to further refine our manuscript. In response to your comments, we have made two main changes in the revised manuscript:

- In response to Editor's comments, we have added comparisons between the ELM simulations and four reference datasets of different variables.
- In response to referee2's comments, we have updated the method section to clarify the purpose of the spatial scaling analysis to avoid misunderstanding.

The point-by-point responses to the specific comments are provided below in blue. All line numbers listed below correspond to those in the clean version of the revised manuscript. We hope that our modifications have addressed all the concerns raised, and we appreciate your consideration of our revised manuscript.

Sincerely,

Lingcheng Li and co-authors

*Editor*

Given the issues raised by the reviewers, I recommend providing authors an opportunity for additional revisions. The authors should focus on addressing major issues regarding the evaluations against observations, which were raised by previous reviews but not well addressed by the authors. The third reviewer also raised an additional concern related the issues of upscaling. Once these concerns are addressed, the manuscript can be evaluated for potential acceptance.

We acknowledge the reviewers' concerns regarding the evaluations against observations. To address this, we have conducted a comprehensive analysis of the ELM simulations with reference datasets, including SM, LH, ELR, and ASR. It is important to note that no model parameter recalibration was conducted (see discussions for details).

Therefore, we added the following content.

In L26–L28,

"The comparison against four benchmark datasets indicates that ELM generally performs well in simulating soil moisture and surface energy fluxes."

In L345–L356,

"**3.4 Reference datasets for evaluating ELM simulation**

We also performed a comparison of all four ELM-simulated variables against reference datasets. It is important to note that we used the default model parameters and did not perform any calibration (see discussions for details). For reference datasets, soil moisture was obtained from the Global Land Evaporation Amsterdam Model (GLEAM; Martens et al., 2017), latent heat flux data was from the MODIS product (Running et al., 2021), and both ELR and ASR data were processed from the land component of the fifth generation of European ReAnalysis (ERA5_Land; Muñoz-Sabater et al., 2021). For the soil moisture evaluation, we compared the surface layer soil moisture from GLEAM (10 cm depth) with the weighted average of the first four-layer soil moisture from ELM (about 11 cm depth). To ensure comparability, we unified the spatial resolution of both reference datasets and ELM simulations to a 0.5-degree resolution and focused our analysis on the annual mean data for 2014."

In L593–L614.

**"4.7 Comparison of ELM simulation against reference data**

The average spatial biases between ELM and reference datasets across CONUS are relatively small, with SM bias at -0.01 m³/m³, LH bias at 1.8 W/m², ELR bias at -3.8 W/m², and ASR bias at 1.1 W/m² (Figure 13 and Figure S27). The correlation coefficient ($R^2$) between ELM and reference datasets was relatively high at 0.60 (for SM), 0.70 (for LH), 0.96 (for ELR), and 0.90 (for ASR). However, the spatial distribution of these biases exhibits variability, with some areas showing more pronounced biases than others. Specifically, in comparison with GLEAM SM, ELM tends to underestimate SM in the southeastern Texas and across the eastern and southeastern CONUS, while it overestimates SM in the western, central, and southwestern CONUS, including the central eastern US which are primarily agricultural areas. For LH, ELM simulates higher values than the MODIS LH dataset in the western and central US and Florida, but lower values in regions such as the eastern and northeastern CONUS, the western US coastal areas, and the Pacific Northwest. Regarding radiation variables, ELM generally underestimates ELR across nearly all of CONUS and tends to overestimate ASR, particularly in the southwestern, southern, eastern, northeastern, and northern regions of CONUS.

[Figure]

Figure 13. Annual mean bias between ELM-simulated variables and reference datasets over CONUS: (a) SM, (b) LH, (c) ELR, and (d) ASR. The negative values indicate lower ELM values compared to the reference data. The inserted histogram plot illustrates the distribution of grid values. For spatial patterns of the reference datasets, refer to Figure S27. The correlation coefficient ($R^2$) between the ELM simulation and the reference dataset is calculated and displayed in the title of each subplot."

In L708–L713.

"This is clearly evidenced in our ELM demonstration simulations, where, despite relatively low CONUS averaged biases for water and energy simulations, the spatial variation in these biases cannot be overlooked, with some regions exhibiting notably larger biases. It is important to emphasize that enhancing model performance requires not just updated input data, but also appropriate calibration of model parameters and faithful model structures to represent various processes."

Figure S27 in the supplementary.

"

[Figure]

Figure S27. The annual mean for reference data of (a) GLEAM SM, (b) MODIS LH, (c) ERA5_Land ELR, and (d) ERA5_Land ASR over CONUS. The inserted histogram plot illustrates the distribution of grid values. "

*Referee 1*

The authors have addressed issues I raised in my previous comments well, though many things can be improved in future edition of data or by further work. I am content with the modifications for publication of this paper.

We greatly appreciate your time and constructive suggestions during the review process, which have substantially improved the manuscript.

*Referee 2*

Very sorry to be late. I reviewed their revised manuscript. Reading their responses for the previous reviews, their revision looks quite defensive (both previous reviewers asked the authors to compare their results to the observations as major issues, but the authors did not do that).

We have added the comparison between ELM simulations and reference data. Please refer to the above response.

I'd like to add one more concern here. In result section (BTW, it's 4th section instead of 3rd), they showed the information loss (reduction of spatial variability) by upscaling. They upscaled the results of the 1 km simulation as well as the 1 km parameter to examine the reduction in spatial variability. Here, the simulation results with upscaled parameters should be used to investigate the spatial variability reduction for simulated variables (e.g., LH). Otherwise, the "critical role of land surface parameters in contribution to the spatial variability of water and energy in land surface simulations" as stated in L599 can be inappropriate. This is because the low-resolution simulation results with upscaled parameters are different from the upscaled simulation results with high-resolution parameters. This concern affects Figures 8 through 12. Therefore, I would like to request them to show at least a CONUS simulation of 0.1 degrees with upscaled parameters and then confirm the simulation result is as similar to the upscaled from the 1km simulation.

Thank you for your insightful comments and for highlighting concerns regarding our spatial scaling analysis method. Our approach draws on the methodology from the well-established study by Vergopolan et al. (2022), which demonstrated significant information loss—on average 48% and up to 80%—when upscaling 30-m SMAP-HydroBlocks soil moisture (SM) data to 1 km resolution. In their study, no simulations were conducted at 1 km to compare with the 1 km SM upscaled from 30 m resolution, and we followed the same approach.

In our analysis, we aim to quantify the extent of spatial variability information retained at 1 km resolution compared to 12 km resolution within each $0.5° \times 0.5°$ box, thereby highlighting the potential benefits of high-resolution simulations. Our analysis should not be interpreted to suggest that low-resolution simulations with upscaled parameters are equivalent to upscaled low-resolution simulations derived from high-resolution model configurations with high-resolution parameters.

We acknowledge that simulations at low resolution using upscaled parameters can yield different spatial variabilities compared to upscaled simulations from high-resolution simulations. Specifically, within each 0.5° × 0.5° box, the comparison of spatial variability in 12 km upscaled SM simulations versus directly simulated SM at 12 km could be different because of the non-linearity of many processes represented by land surface models.

To clarify our methodology and avoid misunderstanding, we have added the following content in section 3.2, L321–325.

"It is crucial to clarify that the upscaled 1 km simulation results in the spatial scaling analysis are not equivalent to the results obtained from a coarse resolution ELM conducted using upscaled parameters. The spatial scaling analysis is intended to emphasize the value of high-resolution modeling in capturing fine-scale spatial variabilities, and to highlight the contributions of high-resolution land surface parameters on the simulated variables."